# Computer prediction and genetic analysis identifies retinoic acid modulation as a driver of conserved longevity pathways in genetically diverse *Caenorhabditis* nematodes

Stephen A Banse[1], Christine A Sedore[1], Anna Coleman-Hulbert[1], Erik Johnson[1], Brian Onken[2], David Hall[3], Erik Segerdell[1], E Grace Jackson[1], Yuhua Song[2], Haley C Osman[3], Jian Xue[2], Elena Basttistoni[3], Suhzen Guo[2], Anna Foulger[3], Madhuri Achanta[2], Mustafa Sheikh[3], Theresa Fitzgibbon[3], John H Willis[1], Gavin C Woodruff[1], Monica Driscoll[2], Gordon Lithgow[3], Patrick C Phillips[1]*

[1]Institute of Ecology and Evolution, University of Oregon, Eugene, United States; [2]Rutgers University, Department of Molecular Biology and Biochemistry, Piscataway, United States; [3]The Buck Institute for Research on Aging, Novato, United States

*For correspondence: pphil@uoregon.edu

Competing interest: The authors declare that no competing interests exist.

## eLife Assessment

This **important** study explores the power of computational methods to predict lifespan-extending small molecules, demonstrating that while these methods significantly increase hit rates, experimental validation remains essential. The study uses all-trans retinoic acid in *Caenorhabditis elegans* as a model, providing genetic and transcriptomic insights into its longevity effects. The data are **compelling** in describing a robust, computationally informed screening process for discovering compounds that extend lifespan in this species.

**Abstract** Discovery of new compounds that ameliorate the negative health impacts of aging promises to be of tremendous benefit across a number of age-based comorbidities. One method to prioritize a testable subset of the nearly infinite universe of potential compounds is to use computational prediction of their likely anti-aging capacity. Here, we present a survey of longevity effects for 16 compounds suggested by a previously published computational prediction set, capitalizing upon the comprehensive, multi-species approach utilized by the *Caenorhabditis* Intervention Testing Program. While 11 compounds (aldosterone, arecoline, bortezomib, dasatinib, decitabine, dexamethasone, erlotinib, everolimus, gefitinib, temsirolimus, and thalidomide) either had no effect on median lifespan or were toxic, 5 compounds (all-trans retinoic acid, berberine, fisetin, propranolol, and ritonavir) extended lifespan in *Caenorhabditis elegans*. These computer predictions yield a remarkable positive hit rate of 30%. Deeper genetic characterization of the longevity effects of one of the most efficacious compounds, the endogenous signaling ligand all-trans retinoic acid (atRA, designated tretinoin in medical products), demonstrated a requirement for the regulatory kinases AKT-1 and AKT-2. While the canonical Akt-target FOXO/DAF-16 was largely dispensable, other conserved Akt-targets (Nrf2/SKN-1 and HSF1/HSF-1), as well as the conserved catalytic subunit of AMPK AAK-2, were all necessary for longevity extension by atRA. Our results highlight the potential of combining computational prediction of longevity interventions with the power of nematode functional genetics and underscore that the manipulation of a conserved metabolic regulatory

circuit by co-opting endogenous signaling molecules is a powerful approach for discovering aging interventions.

## Introduction

Aging is a primary risk factor for a myriad of chronic illnesses, health declines, and mortality. A central premise of research in the current aging field is that aging per se can be treated directly, leading to ancillary benefits across a broad range of age-related comorbidities (the 'geroscience hypothesis'; *Austad, 2016*; *Kennedy et al., 2014*). But how best to identify compounds holding the potential for broad-spectrum effects across an individual's lifespan? While comprehensive screens using model organisms such as the nematode *Caenorhabditis elegans* provide a good approach (*Petrascheck et al., 2007*), a complementary alternative is to use emerging databases of compound-specific physiological effects to predict which compounds are most likely to lead to positive effects on extending lifespan (*Janssens et al., 2019*; *Ribeiro et al., 2023*). An advantage of this approach is that the predictive models should become better and better as the training set of positive hits continues to expand over time (*Vanhaelen et al., 2020*; *Zhavoronkov et al., 2019*). Still, the efficacy of any predictive model is strongly dependent on the quality of the input data, and the well-documented heterogeneity of aging as a phenotype, as well as general challenges in reproducibility per se, create barriers to the successful application of predictive approaches to aging research. The *Caenorhabditis* Intervention Testing Program (CITP) tests compounds for lifespan and healthspan effects across a genetic diversity panel of *Caenorhabditis* nematode strains (*Lucanic et al., 2017*). Beyond *robustness* of response across genetic backgrounds, the CITP has painstakingly focused on *reproducibility* across laboratories and trials via standardization of methods and a hierarchical statistical approach that accounts for experimental variation at a variety of levels of replication. These features make the CITP an ideal framework for testing computer predictions of longevity interventions and serve as the foundation for data collection for improved models in the future.

As a first step toward testing the efficacy of computational prediction of lifespan-extending compounds, we used a previously published set of compound predictions developed via an analysis of the overlap of drug-induced and aging-related gene expression and protein interactions (*Fuentealba et al., 2019*) to develop a list of candidate compounds for further investigation using the CITP workflow. We prioritized compounds with the highest predictive scores and eliminated several compounds whose effects in *C. elegans* were already well characterized. Our analysis led to a set of 16 compounds (aldosterone, all-trans retinoic acid (atRA), arecoline, berberine, bortezomib, dasatinib, decitabine, dexamethasone, erlotinib, everolimus, fisetin, gefitinib, propranolol, ritonavir, temsirolimus, and thalidomide) selected for further testing. As outlined below, we found that of the five candidate compounds – atRA, berberine, fisetin, ritonavir, and propranolol – that extended median lifespan, propranolol and atRA conferred the largest positive effects. Potential confounding interactions of propranolol with the bacterial food of the nematodes led us to focus on atRA for more in-depth genetic and functional analysis.

atRA is an FDA-approved intervention used topically in dermatology and systemically as a chemotherapeutic adjuvant (*Giuli et al., 2020*; *Szymański et al., 2020*). Endogenously, atRA is the most bioactive retinoid derived from vitamin A, known to function as a highly conserved signaling ligand involved in transcriptional regulation (*Albalat, 2009*; *Albalat and Cañestro, 2009*; *Fonseca et al., 2020*). In *C. elegans*, the presence of vitamin A metabolism pathway genes (*Kostrouch et al., 1995*; *Yilmaz and Walhout, 2016*), metabolism of exogenous vitamin A into retinal and atRA (*Chen et al., 2018*), known affinity of *C. elegans* fatty acid- and retinol-binding proteins for retinoids (*Garofalo et al., 2003*), and endogenous atRA detection in untreated animals (*Chen et al., 2018*) combine to suggest the presence of an endogenous nematode atRA signaling pathway. While conservation of the ligand atRA is well supported, the canonical vertebrate downstream retinoid receptors (RXR and RAR) that effect transcriptional responses have not been identified in nematodes. In contrast with the elusive retinoic acid receptors, however, the mammalian kinases modulated by atRA have extensively studied *C. elegans* orthologs. In humans, atRA modulates transcription via PI3K/Akt (*Bastien et al., 2006*; *Ben-Sasson et al., 2011*; *Farias et al., 2005*; *García-Regalado et al., 2013*; *Lee et al., 2014*; *Masiá et al., 2007*; *Qiao et al., 2012*) and p38 MAPK (*Alsayed et al., 2001*; *De Genaro et al., 2013*; *Hormi-Carver et al., 2007*; *Lee et al., 2008*; *Roe et al., 2020*; *Shinozaki et al., 2007*) kinase

signaling. Functionally, kinase signaling is likely mediated by atRA regulation of the kinase phosphorylation state, as has been shown for Akt in mammalian (*Bastien et al., 2006*; *García-Regalado et al., 2013*; *Qiao et al., 2012*) and avian (*Yu et al., 2012*) cell culture.

Building upon our general screening approach, we present a more comprehensive genetic analysis of atRA impact on longevity that suggests functional conservation of atRA kinase regulation, as the effects of atRA on longevity require kinases encoded by both *akt-1* and *akt-2*. In *C. elegans* and mammals, Akt kinases regulate powerful aging pathways (e.g., insulin-like signaling (IIS), FOXO, and Nrf2). Our genetic analysis of atRA longevity in *C. elegans* suggests that the FOXO/DAF-16 transcription factor is not necessary, consistent with atRA acting downstream of, or in parallel to, FOXO. In contrast to FOXO/DAF-16, the Akt-phosphorylation targeted Nrf2 homolog SKN-1 and heat shock transcription factor 1 homolog HSF-1, along with the conserved catalytic subunit of the energy sensor AMPK AAK-2, are required for atRA-induced lifespan extension. The conservation of atRA as a signaling molecule, and the pathways through which atRA affects metabolism and lifespan, anchor the prediction that all-trans retinoic acid intervention (or atRA chemical variants) will translate into efficacious anti-aging in future mammalian and clinical studies.

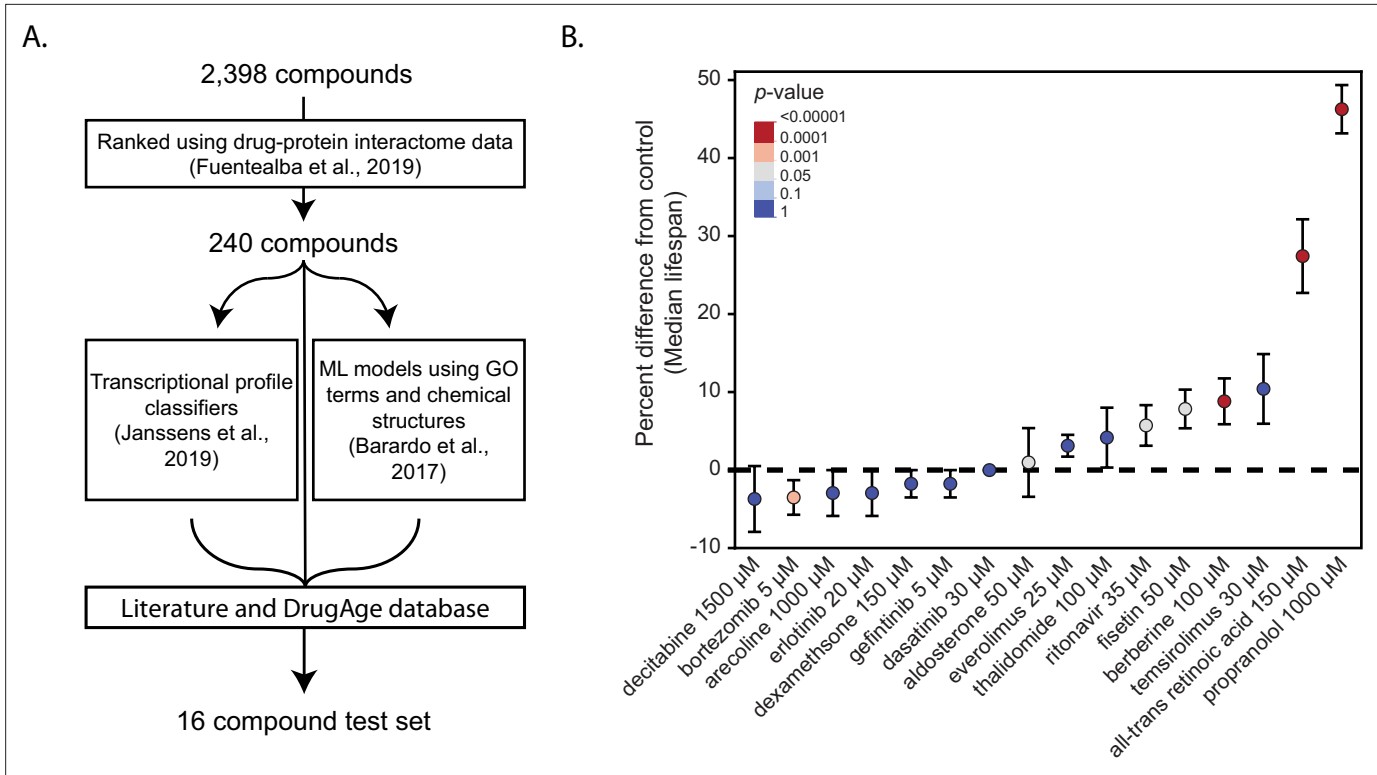

**Figure 1.** Summary of lifespan effects for candidate compounds. (**A**) Compounds were selected for testing by filtering the top 10% of predicted hits from *Fuentealba et al., 2019* and cross-referencing for compounds that also appeared in the top 10% of *Barardo et al., 2017a*, or *Janssens et al., 2019*, or had been shown to work in other model organisms via the DrugAge database. Candidate compounds were then filtered using the DrugAge database and a literature search to deprioritize compounds previously characterized as extending lifespan in *C. elegans* to generate a list of 16 compounds for screening using lifespan analysis. (**B**) Percent difference in median lifespan of individual trial plates compared to the median survival from their pooled carrier control (DMSO and $H_2O$) for animals treated with one of 16 candidate compounds selected for preliminary analysis. The dot represents the mean of all plate replicates across two trials, and the bars represent the standard error. Shown are the results from the longest-lived concentration treatment (4–5 concentrations were tested) for each candidate compound. The shown p-values and error bars are taken from the hierarchical CPH model (see Materials and methods).

The online version of this article includes the following figure supplement(s) for figure 1:

**Figure supplement 1.** Longevity analysis screen in *C. elegans* N2 of 14 candidate compounds that were not selected for further study.

## Results

### Experimental testing of computational predictions identifies all-trans retinoic acid as a candidate pro-longevity intervention

To select compounds for CITP testing (*Figure 1A*), we began with the top 10% of candidates from a published list of computationally ranked compounds built using known drug-protein interactions (*Fuentealba et al., 2019*). To avoid duplicative effort and to favor novel discovery, we used the DrugAge database (*Barardo et al., 2017b*) to de-prioritize compounds that had already been published to extend *C. elegans* lifespan. We then selected 16 candidate interventions by cross-referencing the remaining compounds with the top 10% of two additional computational efforts that predicted aging effects based on comparative transcriptional responses (*Janssens et al., 2019*) (all-trans retinoic acid, arecoline, propranolol, thalidomide) and machine-learning models based on gene ontologies and physical structures (*Barardo et al., 2017a*) (aldosterone, berberine, bortezomib,

**Table 1.** CITP tested compounds that meet computational prediction selection criteria of this study.

| Candidate | Predictions | CITP publication | Beneficial effect on median survival in *C. elegans* in CITP? | Pathway/mode of action |
|---|---|---|---|---|
| Bortezomib | F + B | This study | NP | Proteasome inhibitor (*Chen et al., 2011*) |
| Metformin | F + J | *Onken et al., 2022* | 41% increase at 70 mM | Anti-diabetes |
| 17-Alpha estradiol | F + J | *Banse et al., 2024b* | NP | Estrogen receptor agonist |
| Rapamycin | F + J | *Banse et al., 2024b* | NP | mTOR inhibitor |
| Aspirin | F + J | *Lucanic et al., 2017* | NP | NSAID |
| Imatinib | F + J + B | *Coleman-Hulbert et al., 2019* | NP | Tyrosine kinase inhibitor |
| atRA | F + J | This study | 29% extension at 150 µM | Collagen formation, activates xenobiotic metabolism |
| Dasatinib | F + B | This study | NP | Tyrosine kinase inhibitor |
| Temsirolimus | F + B | This study | NP | mTOR inhibitor |
| Gefitinib | F + B | This study | NP | EGFR inhibitor (tyrosine kinase inhibitor) |
| Resveratrol | F + J | *Lucanic et al., 2017* | 12% extension at 100 µM | Sirtuin activator |
| Berberine | F + B | This study | 16.3% extension at 100 µM | AMPK activator |
| Decitabine | F + B | This study | NP | Nucleic acid synthesis inhibitor |
| Erlotinib | F + B | This study | NP | EGFR inhibitor (tyrosine kinase inhibitor) |
| Valproic acid | F + J | *Lucanic et al., 2017* | NP | Blocks sodium-gated ion channels, increases GABA |
| Aldosterone | F + B | This study | Significant at 50 µM due to late life effects, no change in median lifespan | Steroid hormone |
| Dexamethasone | F + B | This study | NP | Anti-inflammatory corticosteroid |
| Propranolol | F + J | This study | 44% extension at 1 mM* | Beta-blocker |
| Thalidomide | F + J | This study | NP | TNF-a inhibition |
| Arecoline | F + J | This study | NP | Muscarinic agonist (inhibits pharyngeal pumping) |
| Ritonavir | F + B | This study | 4.1% extension at 20 µM | HIV protease inhibitor: inhibits enzymes that normally metabolize other protease inhibitors (primarily in intestines, liver, etc.) |
| Fisetin | F + D | This study | 11.7% extension at 50 µM | Sirtuin activator |
| Everolimus | F + D | This study | NP | mTOR inhibitor |

*May be an indirect effect. NP = no positive effect detected. F – in the top 10% of *Fuentealba et al., 2019*, B – in the top 10% of *Barardo et al., 2017a*, J – in the top 10% of *Janssens et al., 2019*, and D – a positive listing in DrugAge *Barardo et al., 2017b*. Compounds denoted in **bold** were tested by the CITP for this study, while other listed compounds were tested previously by the CITP.

dasatinib, decitabine, dexamethasone, erlotinib, gefitinib, ritonavir, temsirolimus), or listing in the DrugAge database with published lifespan extension in other systems everolimus (*Spindler et al., 2012*) and fisetin (*Yousefzadeh et al., 2018*).

The selected compounds comprise a number of common aging-related classes, including bortezomib (a proteasome inhibitor; *Chen et al., 2011*), fisetin (a sirtuin activator; *Kim et al., 2015*), temsirolimus (a PI3K/mTOR inhibitor and derivative of rapamycin; *Ali et al., 2022*), and dasatinib (a tyrosine kinase inhibitor; *Talpaz et al., 2006*), among others (*Table 1*). We then screened the selected compounds at 4–5 concentrations using full lifespan analysis (*Figure 1—figure supplement 1*). Among the 16 candidate interventions, 3 were water-soluble and 13 were DMSO-soluble. While DMSO can impact lifespan (*Wang et al., 2010*), we did not observe a difference between the $H_2O$ and DMSO vehicle control treatments (median lifespan 17 days for both, p = n.s.), consistent with the published absence of DMSO effects at concentrations similar to those used in our studies (*AlOkda and Van Raamsdonk, 2022*). Among the 16 computationally prioritized candidate compounds, we found that aldosterone, dexamethasone, erlotinib, decitabine, dasatinib, everolimus, thalidomide, and temsirolimus did not significantly change median lifespan at any tested concentration (*Figure 1B*; *Figure 1—figure supplement 1*).

Among the eight remaining candidates, we found that three compounds shortened median lifespan (arecoline, gefitinib, and bortezomib). Tests of muscarinic/nicotinic agonist arecoline at five concentrations ranging from 50 µM to 8 mM revealed toxic effects at the highest concentration (–29.4% median lifespan, p < 0.0001). The epidermal growth factor inhibitor gefitinib also had small, but significant, negative effects at 10, 25, 50, and 80 µM (*Figure 1—figure supplement 1*). In contrast, the 26S proteasome inhibitor bortezomib conferred strong toxicity effects that increased with concentration through the entire concentration range we tested (5, 10, 20, and 30 µM, –10.5% to –47.3% median lifespan; p = 0.0002 at 5 µM and p < 0.0001 at all other concentrations; *Figure 1—figure supplement 1*). Thus, some compounds computationally predicted to enhance longevity can be found to be toxic when empirically investigated, underscoring that validation is a key element of any prediction pipeline.

We found that the remaining five compounds conferred statistically significant positive effects on median lifespan for at least one tested concentration (*Figure 1B*; *Figure 1—figure supplement 1*), representing a hit success rate of 31.25% (5/16) for our test set (*Table 1*). Seven additional compounds met our selection criteria, but because they had been previously tested by the CITP, we did not include these compounds in tests presented here (see *Table 1*). With those previously tested compounds included, we see a similar overall hit success rate of ~30% (7/23) (*Table 1*). The bioactivity of these compounds was as follows: ritonavir had effects at 20 and 35 µM (6.3%, p = 0.0016 and p = 0.0282, respectively). The sirtuin activation/mTOR inhibitor fisetin had positive effects at 10, 50, and 100 µM (p = 0.0081, p = 0.0025, and p = 0.0032, respectively), with an effect size up to 11.8%, and no effect detected at 20 µM. The AMPK activator berberine conferred significant effects only at 100 µM, with an 11.8% increase in median lifespan (p < 0.0001). In support of the potential translatability of the computationally predicted candidate compounds tested here, fisetin (*Yousefzadeh et al., 2018*) and berberine (*Dang et al., 2020*) have also been found to increase median lifespan in mice.

The two remaining candidate interventions induced large increases in longevity, with propranolol extending median lifespan 44.4% at 1 mM (p < 0.0001) and atRA extending median lifespan 23.5% at 150 µM (p < 0.0001) (*Figure 1B*). Propranolol and atRA are particularly interesting interventions because they are both FDA-approved drugs, potentially providing an easier path toward clinical use as aging interventions. For example, propranolol is a well-tolerated drug with a long history of use (*Srinivasan, 2019*, p. 50; *Zacharias et al., 1972*) treating high blood pressure (*Prichard and Gillam, 1964*), angina (*Hamer et al., 1964*), and atrial fibrillation (*Rowlands et al., 1965*). While we observed a large (44.4%, p < 0.0001) median lifespan extension at 1 mM propranolol, we also saw extension at 0.5 mM (16.7%, p < 0.0001) (*Figure 2D*). In contrast to the positive effects that we observed at 0.5 and 1 mM, we found that increasing the treatment concentration to 5 mM propranolol resulted in toxicity and a reduction in lifespan (–61.1%, p < 0.0001) (*Figure 2D*). These observations led us to extend our tests into the related species *C. briggsae* (AF16) and *C. tropicalis* (JU1630). In these two species, we observed similar toxicity at 5 mM (p < 0.0001), but no beneficial effects at lower concentrations (*Figure 2E, F*).

In human applications, propranolol functions as a general antagonist of β1 and β2 beta-adrenergic receptors. We therefore sought to determine if a β1-specific antagonist like metoprolol could

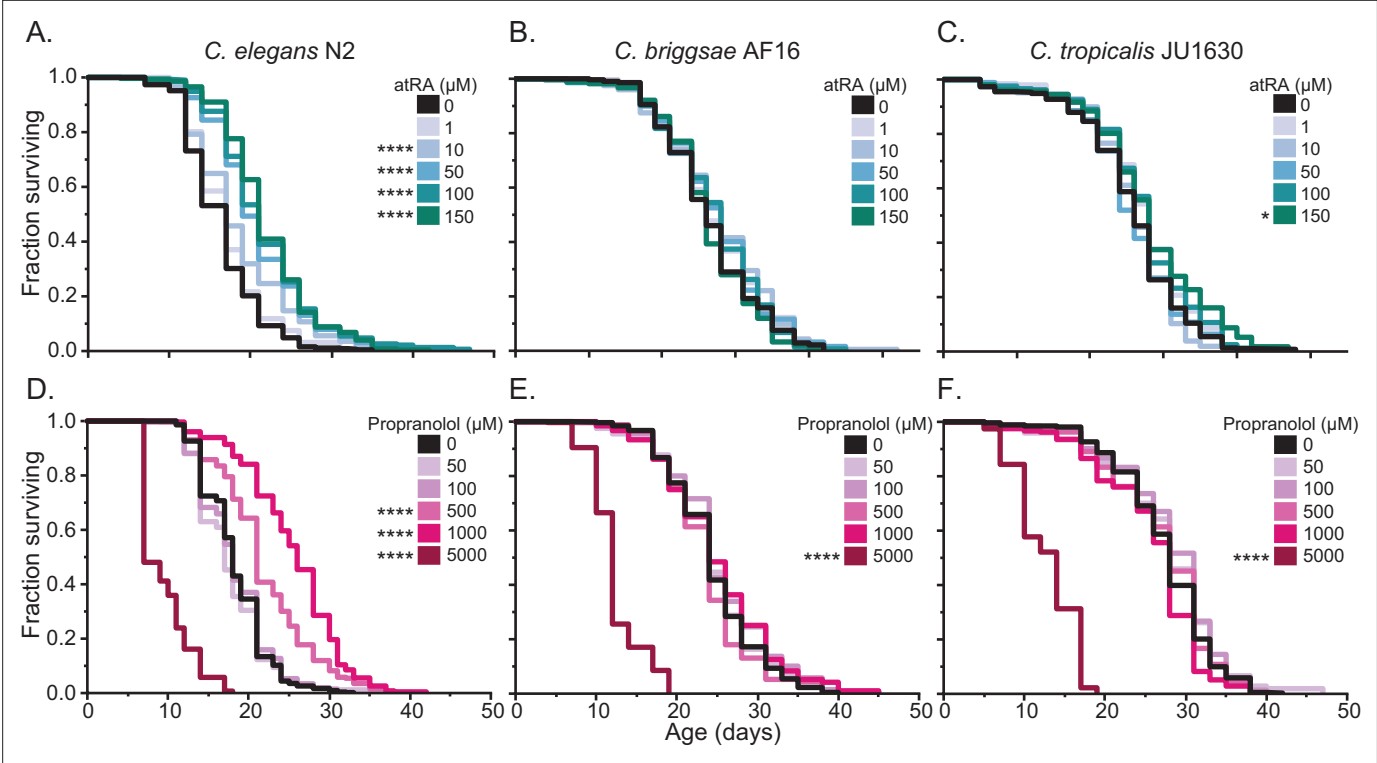

**Figure 2.** Dose-dependent lifespan effects of all-trans retinoic acid and propranolol across diverse *Caenorhabditis* species. (**A–C**) Manual lifespan analysis of five concentrations of atRA (black – DMSO control, increasing levels of teal 1–150 µM) and (**D–F**) propranolol (black – H₂O control, increasing levels of pink – 50–5000 µM) on three *Caenorhabditis* species. The upper limit tested was determined by compound solubility (atRA), or toxicity (propranolol). The Kaplan–Meier curves presented consist of pooled replicates from two trials. Asterisks represent p-values from the CPH model such that ****p < 0.0001, ***p < 0.001, **p < 0.01, and *p < 0.05.

The online version of this article includes the following figure supplement(s) for figure 2:

**Figure supplement 1.** Median lifespan dose response of all-trans retinoic acid and propranolol in *C. elegans* N2.

**Figure supplement 2.** Metoprolol does not extend lifespan in *C. elegans*, nor does propranolol on PFA-treated OP50-1, consistent with its bacteriostatic activity.

recapitulate the longevity effects in *C. elegans*. We assayed longevity effects for metoprolol across a concentration range of 5 µM to 1.5 mM and observed no positive effects (*Figure 2—figure supplement 2A*). Although there could be multiple reasons that metoprolol was not effective, we followed up by asking whether the effects of propranolol require β2 antagonism or are unrelated to β-adrenergic antagonism. A β-adrenergic-independent mechanism was suggested by the change we noted in the appearance of the bacterial lawns on propranolol-treated plates. When we tested bacterial growth, we found that propranolol reduced bacterial growth at the same concentrations at which we observed lifespan effects (*Figure 2—figure supplement 2C*). The propranolol impact on bacterial food source growth suggests a potential indirect food-dependent mechanism for propranolol on *C. elegans* lifespan. We therefore repeated the lifespan studies at 0.5 and 1 mM in the presence of paraformaldehyde-treated bacteria that are metabolically inert (*Beydoun et al., 2021*). Under conditions in which propranolol effects on bacterial growth were eliminated, we observed shorter lifespans in populations treated by propranolol (*Figure 2—figure supplement 2B*). These observations suggest that propranolol either does not exert direct beneficial effects on lifespan in *C. elegans* or has confounding direct and indirect effects that depend on bacterial food state. The potential food-dependent effects of propranolol require further study beyond the scope of our current screening protocols. Therefore, we elected to focus on the large lifespan extension generated by atRA treatment for the remainder of this study.

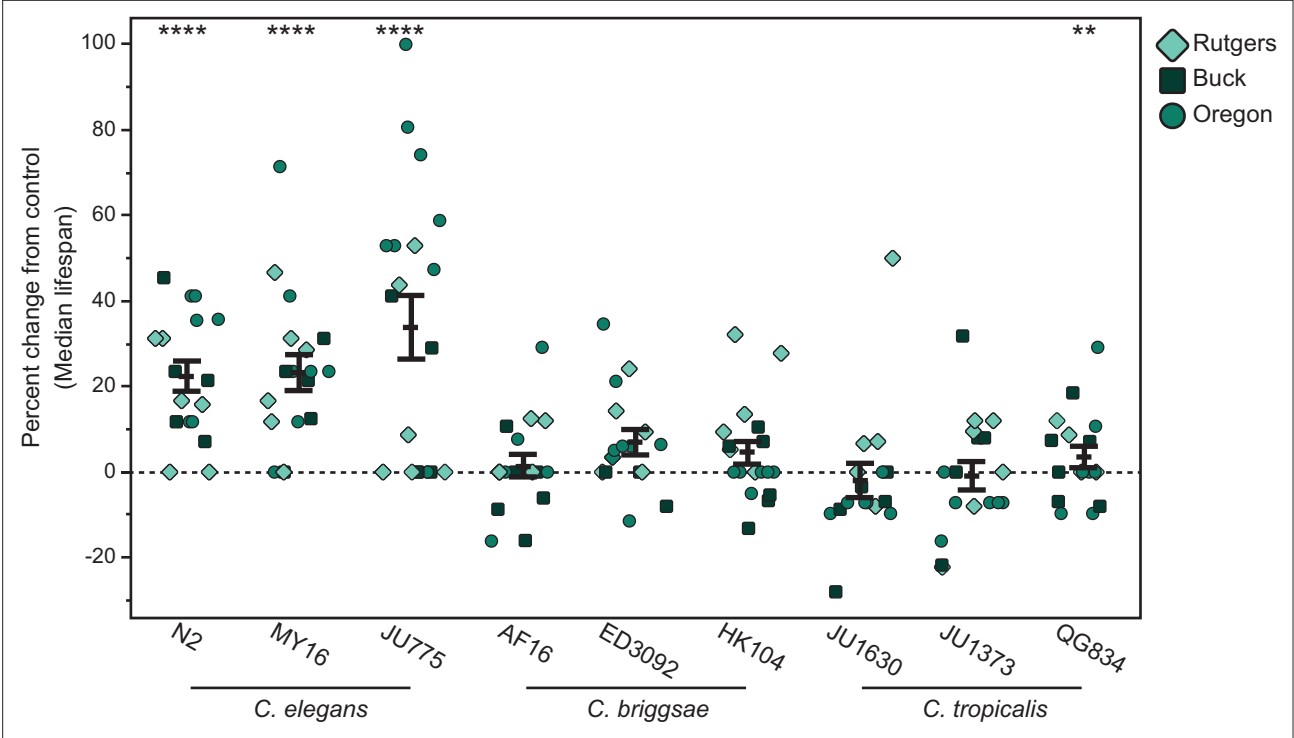

**Figure 3.** The vitamin A derivative atRA extends life in a species-specific manner. The effect of adult exposure to 150 µM atRA on median survival in manual lifespan assays. Three strains were tested from each of three species: *C. elegans* strains N2, JU775, and MY16, *C. briggsae* AF16, ED3092, and HK104, and *C. tropicalis* strains JU1630, JU1373, and QG834. Each point represents the percent change in median survival for an individual trial plate relative to the vehicle control median. The bars represent the mean ± the standard error of the mean. Replicates were completed at the three CITP testing sites (circles – Oregon, squares – Buck Institute, and diamonds – Rutgers). Error bars and asterisks represent p-values from the hierarchical CPH model such that ****p < 0.0001, ***p < 0.001, **p < 0.01, and *p < 0.05.

The online version of this article includes the following figure supplement(s) for figure 3:

**Figure supplement 1.** Lifespan curves for the data presented in *Figure 3* – manual survival of nine *Caenorhabditis* strains with 150 µM atRA treatment.

**Figure supplement 2.** Survival of nine CITP strains with atRA treatment on the ALM.

**Figure supplement 3.** Swimming ability of the nine CITP strains with atRA treatment.

## Longevity extension via atRA treatment is dependent upon genetic background

The scientific literature is rife with examples of intervention effects on longevity that vary in response to experimental differences. Indeed, previous experiments treating *C. elegans* with atRA have resulted in contradictory effects (*Janssens et al., 2019*; *Statzer et al., 2021*) for reasons that are not entirely clear. To determine the most efficacious concentration of atRA treatment, we tested a dosage range from 1 to 150 µM. For *C. elegans* N2, we observed increasing positive effects at all tested concentrations above 1 µM (which had no detectable effect; *Figure 2*) and a very slight but significant increase in median lifespan for *C. tropicalis* JU1630 at 150 µM atRA (7.7%, p = 0.0215). However, we observed no effects in *C. briggsae* across the tested concentration range (*Figure 2B*). Given these observations, we elected to use 150 µM atRA for the remainder of the experiments in this study. In contrast with propranolol, we observed no obvious effects of atRA on bacterial growth using replica plating.

Expanding this analysis across a more extensive genetic diversity set following the full CITP replication protocol, we tested the effects of atRA on three strains of *C. elegans* (N2, JU775, MY16), *C. briggsae* (AF16, ED3092, HK104), and *C. tropicalis* (JU1630, JU1373, QG834) (*Figure 3*) with replication at three distinct geographic sites (University of Oregon, Rutgers University, and the Buck Institute). Total genetic variation across *C. elegans* strains is comparable to that observed among humans, while the differences among species are comparable to the genetic distance between humans and mice (*Teterina et al., 2022*). We found that the substantial atRA-associated effects on longevity are

robust to genetic variation across all three *C. elegans* strains, yielding lifespan extensions of 18.8–44.4% (*Figure 3*; *Figure 3—figure supplement 1A*). While the lifespan extension initially observed in *C. tropicalis* JU1630 failed to replicate, *C. tropicalis* QG834 displayed a small but significant increase in lifespan (*Figure 3*; *Figure 3—figure supplement 1C*). Clearly, given the small effect size, we are at the edge of statistical power to detect a positive effect within this species. Again, atRA did not register any significant positive effects in the *C. briggsae* strains (*Figure 3*; *Figure 3—figure supplement 1B*). Partitioning total variation across this large set of experimental replicates, we found only a small amount of variability attributable to site (3.9%) or differences among experimenters (7.1%), with the majority of variance being attributable to individual variation (53.2%; *Supplementary file 1a*), consistent with previous CITP studies (*Banse et al., 2024a*; *Lucanic et al., 2017*). Thus, atRA treatment is reproducible within and between laboratories, but subject to high levels of individual variation, as are all longevity studies.

To increase the temporal resolution of our survivorship curves, we repeated our longevity analyses using the Automated Lifespan Machine (ALM) technology (*Stroustrup et al., 2013*) across the same nine genetically diverse strains. We again observed a positive effect for the three *C. elegans* strains, demonstrating the robustness of atRA longevity effects across genetic backgrounds in these strains (*Figure 3—figure supplement 2*). Interestingly, we do not see any positive effects of atRA on the ALM for *C. briggsae* or *C. tropicalis*, but instead see slightly toxic effects for *C. briggsae* strains AF16 and HK104 and *C. tropicalis* strain JU1630. It is not clear what might drive this difference, although the ALM introduces some different environmental stresses and conditions as compared to manual assay conditions (e.g., repeated light exposure and distinct compound introduction; see *Banse et al., 2019* for discussion).

## atRA tends to enhance locomotory healthspan in *C. elegans*, but not in *C. briggsae* or *C. tropicalis*

A goal of longevity interventions is to enhance physiological health, which, like in humans, can be measured as improvement in older age locomotory capacity. We therefore determined the effect of atRA exposure on aging adult swim performance using a video analysis of swimming behavior (*Ibáñez-Ventoso et al., 2016*; *Restif et al., 2014*). In previous work, we reported that anti-aging interventions can have disparate effects on longevity and adult swimming ability and that treated strains can show positive effects in motility enhancement (*Banse et al., 2024b*) even in the absence of longevity enhancement. Using strain-specific models for swimming behavior to generate a composite swimming score based on eight underlying measures (*Banse et al., 2024b*), we observed significant improvement for two of the *C. elegans* strains at day 12 of adulthood (*Figure 3—figure supplement 3*). Similar to atRA effects on longevity, we find that atRA was generally ineffectual at promoting swimming health in *C. briggsae* and *C. tropicalis*, with improvements only seen in day 16 of AF16 (41.6%, p = 0.00191), while decreased swimming scores were observed in all three *C. tropicalis* strains at one or more test days. Overall, then, atRA has largely positive effects on *C. elegans* longevity and health while it has the potential to be detrimental to *C. briggsae* and *C. tropicalis* depending on the assay type and particular genetic background.

## atRA lifespan extension requires atRA-modulated kinases AKT-1, AKT-2, and AMPK

Given the plasticity of genetically determined longevity within *C. elegans*, we next sought to identify the pathways required for atRA lifespan extension. Because no retinoic acid binding transcription factors have been identified in *C. elegans*, we looked to the known effects of atRA in modulating human kinase activity (*Albalat, 2009*; *Albalat and Cañestro, 2009*; *Fonseca et al., 2020*) to identify candidate pathways. The vertebrate atRA-responsive kinases do have extensively studied orthologs in *C. elegans*. Akt homologs emerged as particularly relevant due to their involvement in longevity-related IIS signaling, and the fact that in mammals, phosphorylation of Akt in response to atRA occurs at a site that appears to be conserved in the *C. elegans* Akt homologs (*García-Regalado et al., 2013*). We therefore asked whether either *akt-1* or *akt-2* was required for atRA lifespan effects by measuring the lifespans of *akt-1(ok525)* and *akt-2(ok393)* loss of function mutants (*Figure 4A, B*). Consistent with previously published studies (*Newell Stamper et al., 2018*; *Soukas et al., 2009*), in control-treated animals, we observed longer median lifespans for the mutants (median 26 and 23 days vs. 17 in

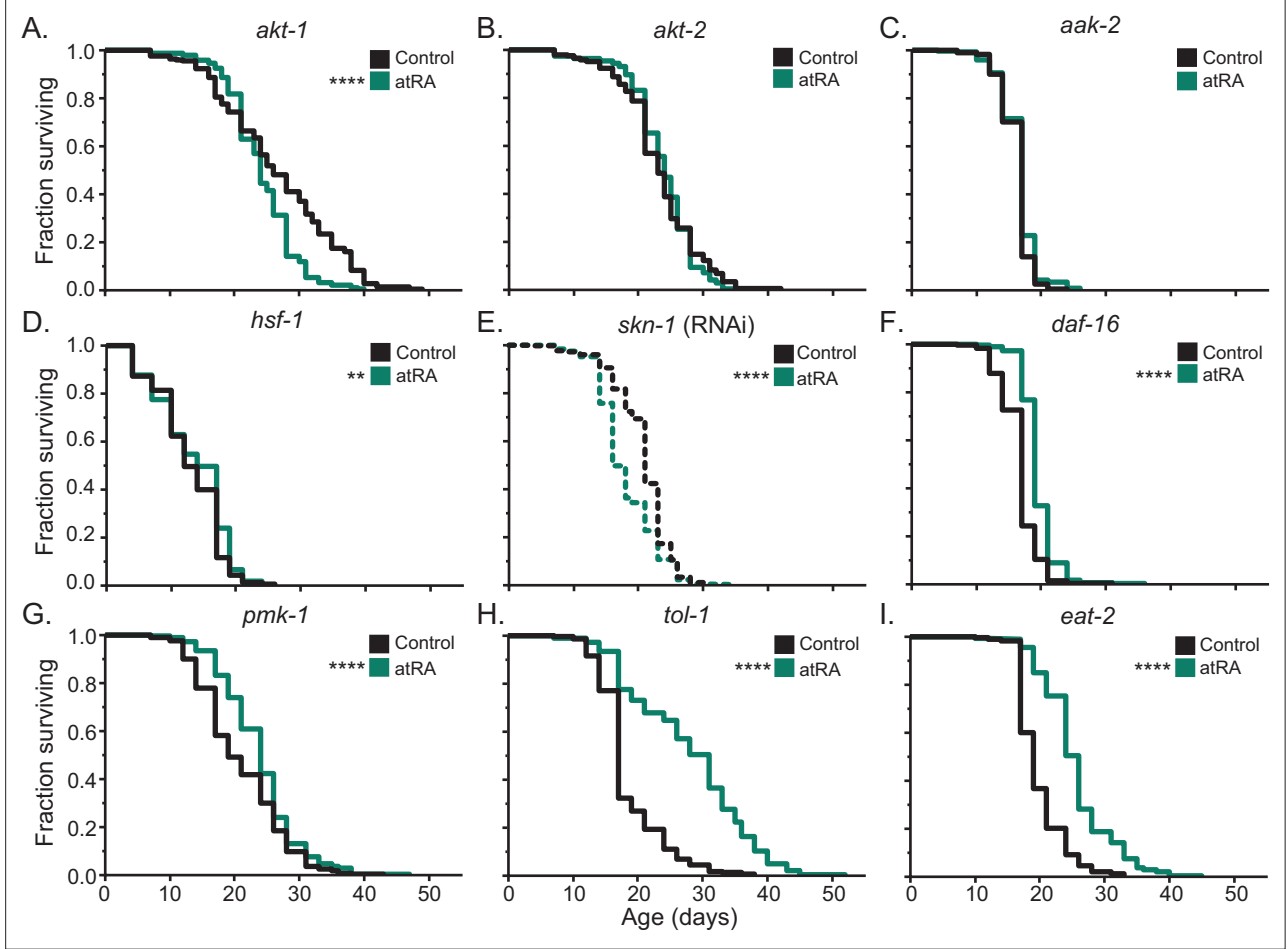

**Figure 4.** Genetic analysis of atRA effects on lifespan. Lifespan analysis under 150 µM atRA (green) or vehicle control (black). For wildtype, the canonical CITP N2 strain was used. This response was compared to loss of function/downregulation of the Akt/protein kinase B (PKB) homologs (**A**) *akt-1* and (**B**) *akt-2*, (**C**) the AMP-activated protein kinase *aak-2*, (**D**) the heat shock transcription factor homolog *hsf-1*, (**E**) the Nrf transcription factor homolog *skn-1*, (**F**) the FOXO transcription factor homolog *daf-16*, (**G**) the p38 MAP kinase homolog *pmk-1*, (**H**) the toll-like receptor *tol-1*, and (**I**) the acetylcholine receptor *eat-2*. For *skn-1*, RNAi knockdown was used because of the lethality of the mutant (see *Figure 4—figure supplement 1B* for control RNAi experiment). For all other genes, loss of function mutants were used. Kaplan–Meier curves include pooled replicates from two trials, except for the RNAi experiment, which consisted of three trials. Asterisks represent p-values from the CPH model such that ****p < 0.0001, ***p < 0.001, **p < 0.01, and *p < 0.05.

The online version of this article includes the following figure supplement(s) for figure 4:

**Figure supplement 1.** SKN-1 regulation and *skn-1* RNAi lifespan.

WT). When we performed longevity analysis of *akt-1(ok525)* in the presence of atRA (*Figure 4A*), we observed a significant decrease in longevity (–7.7% median lifespan, p = 0.00000142), demonstrating a requirement for AKT-1 in atRA-induced longevity extension. Repeating the analysis in *akt-2(ok393)* mutants (*Figure 4B*) demonstrated a complete dependence on AKT-2, with atRA having no significant effect in the *akt-2* mutant background (p = 0.7170). We conclude that atRA longevity effects require *akt-1* and *akt-2*, consistent with a known atRA signaling mechanism in mammals.

Similarly, in human cell culture (*Ishijima et al., 2015*) and mouse models (*Zhang et al., 2019*), retinoic acid activates AMPK. AMPK is a conserved sensor of intracellular energy state that regulates glucose and lipid metabolism. Previous work has shown that AMPK is involved in the transition to gluconeogenesis in the long-lived dauer stage (*Penkov et al., 2020*) and may also play a role in gluconeogenesis in adults (*Nguyen et al., 2020*). In *C. elegans*, AMPK is required for many longevity interventions (*Jayarathne et al., 2020*; *Onken and Driscoll, 2010*; *Peng et al., 2019*), while overexpression of the AMPK catalytic subunit *aak-2* (*Apfeld et al., 2004*), or expression of a constitutively activated AAK-2 (*Greer et al., 2007*; *Mair et al., 2011*), can directly increase lifespan. Additionally,

AMPK both regulates and is regulated by Akt, making AMPK an interesting candidate for involvement in atRA longevity effects. We therefore tested aak-2(ok524) mutants for atRA lifespan extension. We observed that the atRA lifespan extension was fully dependent on aak-2 (*Figure 4C*). Overall, our data suggest that atRA longevity may be mediated through a conserved signaling process.

## Robust atRA lifespan extension requires the HSF-1 and SKN-1 transcription factors

Another set of atRA transducing factors signaling are *hsf-1*, the human heat shock factor 1 transcription factor homolog, and *skn-1*, the nematode homolog of the mammalian Akt-target Nrf2. Previous work has demonstrated that HSF-1 is required for several lifespan-extending genetic and pharmacological interventions in *C. elegans* (*Lazaro-Pena et al., 2022*; *Steinkraus et al., 2008*; *Todorova et al., 2023*) and that skn-1 can directly promote lifespan (*Tang and Choe, 2015*; *Tullet et al., 2017*), as well as being implicated in multiple longevity interventions (*Duangjan et al., 2019*; *Seo et al., 2015Seo et al., 2015*) including vitamin D (*Mark et al., 2016*) and thioflavin T (*Alavez et al., 2011*; *Lucanic et al., 2017*). In mammalian studies, Akt directly regulates HSF1 through phosphorylation (*Carpenter et al., 2015*; *Da Costa et al., 2020*; *Lu et al., 2022Lu et al., 2022*; *Tang et al., 2020*), and hsf-1 has been implicated as a downstream effector of PI3K/Akt signaling that functions in conjunction with DAF-16 to regulate lifespan in *C. elegans* (*Chiang et al., 2012*; *Hsu et al., 2003*). We tested for atRA lifespan extension in a hsf-1(sy441) mutant encoding a premature stop codon that removes the conserved transactivation domain (*Hajdu-Cronin et al., 2004*). In this *hsf-1* background, atRA treatment had a greatly reduced impact on lifespan, showing only a small increase in median lifespan and no extension in maximum lifespan (*Figure 4D*).

Because *skn-1* is an essential gene required developmentally to specify mesodermal fates (*Bowerman et al., 1992*; *Maduro et al., 2001*), we tested for longevity effects of atRA in animals fed HT115 *E. coli* carrying an RNAi vector targeting *skn-1* (*Kamath and Ahringer, 2003*) starting from the L3/L4 developmental stage (skn-1 is an essential gene for early development, and thus assaying a knockout mutation or beginning with earlier interventions is not possible). It should be noted that we observed that the HT115 strain of *E. coli* itself extends lifespan relative to strain OP50, consistent with previous findings (*Stuhr and Curran, 2020*; *Figure 4—figure supplement 1B*). When we compared skn-1 RNAi-treated animals exposed to atRA versus vehicle control, in contrast to the lifespan extension for animals under control RNAi conditions, we observed a 23.8% decrease in median lifespan (p < 0.001) (*Figure 4E* and *Figure 4—figure supplement 1B*), suggesting that atRA is toxic in the absence of skn-1 function. Our data are consistent with hsf-1 and skn-1 being necessary for the lifespan-extending transcriptional response to atRA and/or for addressing potential toxic side effects of atRA.

## The FoxO/DAF-16 transcription factor is not essential for atRA lifespan extension

Given the dependence on *akt-1/2*, we sought to determine if atRA lifespan extension requires the canonical *C. elegans* AKT-target *daf-16*/FOXO, a known regulator of aging (*Murphy and Hu, 2013*) for which activation is a common feature of chemical interventions that extend *C. elegans* lifespan (*Kim et al., 2019*; *Wang et al., 2015*; *Zhao et al., 2017*) (although *daf-16* independent lifespan extension is also possible; *Onken and Driscoll, 2010*). We therefore measured longevity in *daf-16(mu86)* null mutants treated with atRA. We found that *daf-16(mu86)* animals still exhibited a lifespan extension (12%, p < 0.0001) compared to vehicle control animals (*Figure 4F*). The fact that the atRA longevity effect size is larger in wildtype animals (24% vs. 12%) reveals that although *daf-16* contributes in part to the atRA effect, DAF-16 is not absolutely required, and therefore additional or alternative outputs must be operative. Inputs to longevity pathways are well documented to be complex and inter-related (*Narasimhan et al., 2009*; *Nikoletopoulou et al., 2014*; *Parkhitko et al., 2020*). For example, *akt-1* and *akt-2* are primary upstream modulators of *daf-16* in the IIS pathway regulation of the long-lived alternative dauer larval state (*Paradis and Ruvkun, 1998*), but have little effect on IIS modulation of adult longevity, when *sgk-1* becomes the primary regulator of DAF-16 (*Hertweck et al., 2004*). We conclude that atRA acts in part via DAF-16 but infer that atRA either acts independently of the IIS pathway, or primarily through the PI3K/Akt portion of the IIS pathway, which would be consistent with Akt-dependent atRA signaling in mammals (*Bastien et al., 2006*; *García-Regalado et al., 2013*; *Qiao et al., 2012*).

## atRA lifespan extension in *tol-1* and *pmk-1* mutants

While our observation that atRA requires AKT-1/2 and SKN-1 is consistent with a simple signaling cascade in which atRA modulates Akt regulation of SKN-1, more complicated responses are possible. Previous research has established that there is crosstalk between Akt and p38 MAPK signaling in humans (*Gonzalez et al., 2004*) and in *C. elegans* (*Tullet et al., 2008*). Genetic and biochemical analysis of SKN-1 has shown that in addition to Akt regulation, SKN-1 is also post-translationally regulated by the *pmk-1*/p38 MAPK pathway (*Figure 4—figure supplement 1A*). Importantly, work in mammals has implicated p38/MAPK signaling in atRA responses (*Alsayed et al., 2001*; *De Genaro et al., 2013*; *Hormi-Carver et al., 2007*; *Lee et al., 2008*; *Roe et al., 2020*; *Shinozaki et al., 2007*), suggesting that atRA may affect two different signaling cascades that can regulate SKN-1.

In light of these considerations, we addressed p38 MAPK signaling as a potential effector pathway for atRA. *C. elegans* has three known p38 mitogen-activated protein kinase homologs, *pmk-1*, *pmk-2*, and *pmk-3*. SKN-1 is regulated by the MAPK cascade that culminates with p38/PMK-1 phosphorylation of SKN-1 at serines 164 and 430 (*Figure 4—figure supplement 1A*). The phosphorylation of S164 and A430 sites results in increased nuclear SKN-1 levels, resulting in transcription of innate immunity and oxidative stress genes (*Inoue et al., 2005*). When we tested *pmk-1(km25)* mutants for atRA longevity effects, we found that *pmk-1(km25)* mutants exhibit an atRA-induced extension in median lifespan (26.3%, $P<0.0001$), but did not exert an effect on maximum lifespan, suggesting enhanced importance of *pmk-1* later in life (*Figure 4G*). Although our data identify AKT-1 and AKT-2 as more impactful than p38/PMK-1 in atRA-mediated longevity, additive and more complex atRA regulation of SKN-1 by PI3K/Akt and p38 MAPK pathways may be possible. For example, Akt regulates SKN-1 through phosphorylation of serine 12 (*Blackwell et al., 2015*), while *pmk-1* regulates SKN-1 through serines 164 and 430 (*Figure 4—figure supplement 1A*).

To probe the candidate signaling pathways further, we considered potential pathway receptors. *C. elegans PMK-1* functions downstream of *TIR-1* (*Liberati et al., 2004*; *Peterson et al., 2022*), one of two Toll/interleukin-1 receptor homology (TIR) domain-containing genes (*Paysan-Lafosse et al., 2023*). The second TIR-domain containing protein is the membrane-associated TOL-1, which signals through a p38 MAPK cascade including *mom-3* and *pmk-3*, and ultimately IKB-1. We measured the lifespan of *tol-1(nr2033)* mutants treated with atRA to find that *tol-1(nr2033)* animals exhibit an enhanced response to atRA, with an 82.4% increase in median lifespan in the mutant background relative to the 23.5% increase observed in the N2 wildtype background (*Figure 4H*).

## atRA can extend lifespan in a genetic caloric restriction model

One widely conserved mechanism for lifespan extension is caloric restriction. In *C. elegans* longevity research, one frequently used caloric restriction model is genetic mutation of *eat-2*. EAT-2 is a nicotinic acetylcholine receptor expressed in the pharyngeal muscle that facilitates normal, fast feeding behavior (*Avery, 1993*; *McKay et al., 2004*; *Raizen et al., 1995*). In an *eat-2* mutant background, feeding behavior is slowed, inducing a caloric restriction state that extends life (*Lakowski and Hekimi, 1998*), either via dietary restriction itself or via a combination of dietary restriction and an innate immunity response to altered bacterial processing (*Kumar et al., 2019*). Previous work has shown that some compound interventions are incapable of further prolonging *eat-2* lifespan (e.g., metformin; *Onken and Driscoll, 2010*), while other interventions appear independent/additive (e.g., Sonneradon A; *Jiang et al., 2022*) with *eat-2* effects. We were particularly interested in the possibility that atRA might act as an *eat-2*-like dietary restriction mimetic because previous characterization demonstrated that *eat-2* longevity was independent of *daf-16* (*Lakowski and Hekimi, 1998*), but dependent on *skn-1* (*Park et al., 2010*), mimicking our observations for atRA. We therefore treated *eat-2(ad1113)* mutants with vehicle and atRA. Consistent with atRA acting through a mechanism distinct from *eat-2*, we observed a significant atRA-induced extension in lifespan in the *eat-2(ad1113)* animals (36.8% increase in median survival, $p < 2e-16$) (*Figure 4I*). In *C. elegans* studies, caloric restriction can be induced through several different experimental regimes, each of which requires a different set of genetic pathways to exert longevity effects (*Greer and Brunet, 2009*). We therefore conclude that atRA longevity effects are either unrelated to and/or are mechanistically distinct from *eat-2* effects on longevity.

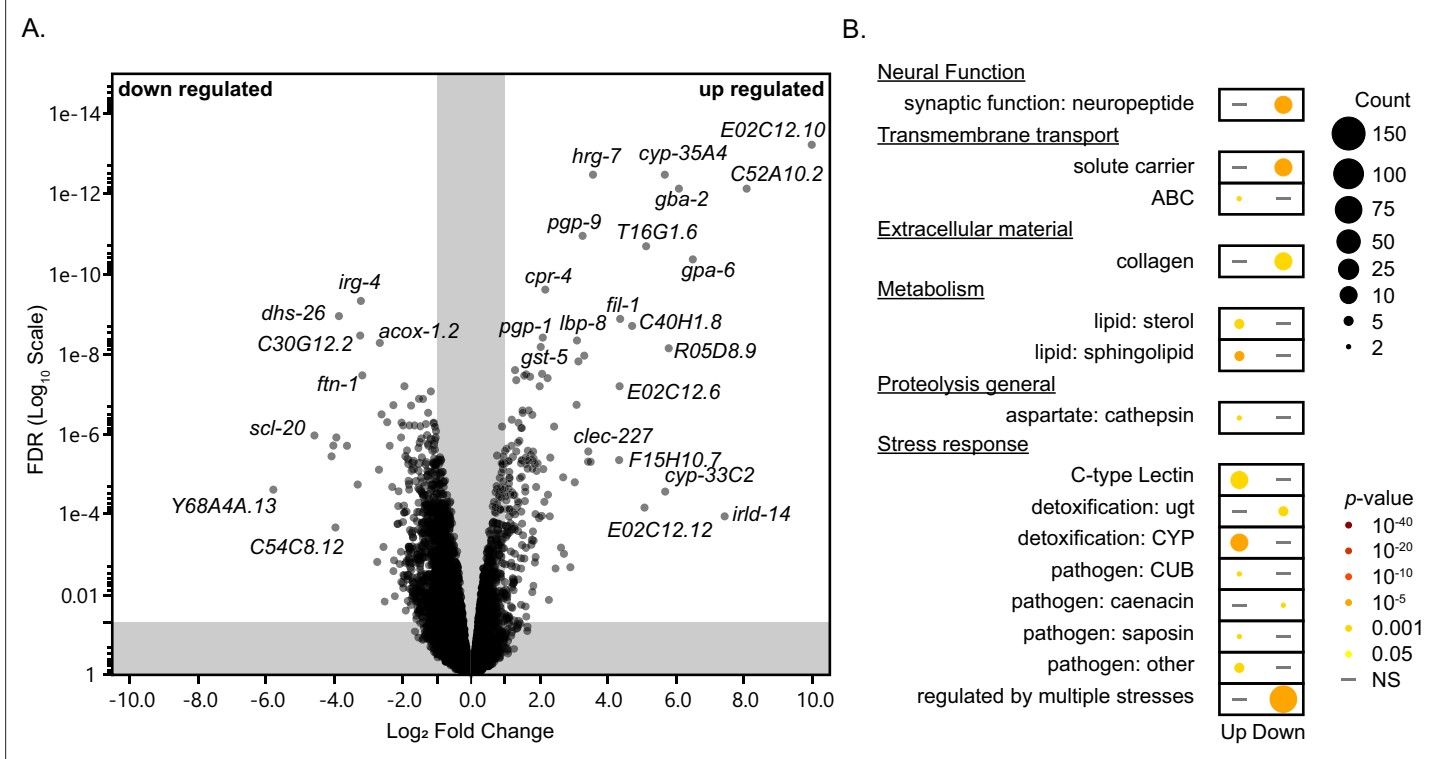

**Figure 5.** Altered transcriptome under atRA treatment. (**A**) Volcano plot for gene expression from RNA-seq experiments performed on day 4 of adulthood exposed to 150 µM atRA or vehicle control. (**B**) Enrichment analysis using WormCat (*Higgins et al., 2022*; *Holdorf et al., 2020*).

The online version of this article includes the following figure supplement(s) for figure 5:

**Figure supplement 1.** Collagens are downregulated in response to atRA exposure.

**Figure supplement 2.** Effects of atRA on transcription of sphingolipid metabolism genes.

## atRA treatment alters gene expression in stress-response pathways

Given the dependency of lifespan extension under atRA treatment, we used RNA-seq to assess transcriptional changes under atRA treatment in wildtype N2 animals. We performed RNA-seq on day 4 adult animals treated with atRA compared to carrier control. We were able to detect the expression of 12,746 *C. elegans* genes in our dataset (*Figure 5A*). Among the detected genes, 17% (2169) were differentially expressed with atRA treatment (FDR <0.05). We defined the subset of differentially expressed genes with an absolute log$_2$ fold change (LFC) greater than one as wildtype atRA response genes (WaRGs). The WaRGs represent ~5.1% of all detected genes (653 total) and were more heavily weighted toward downregulated genes, with 487 (3.8% of total) downregulated versus 166 (1.3% of total) upregulated. Analysis of the expression pattern of the WaRGs shows a skewed distribution among the upregulated genes, with 86% (138/160; $q = 5.9e-38$) of the genes with characterized expression being produced in the intestine. The downregulated genes are enriched for genes expressed in the excretory duct (32/470; $q = 4.1e-9$), excretory socket cell (29/470; $q = 5.1e-08$), and the epithelial system (281/470; $q = 2.7e-8$). Potentially relevant to metabolic regulation of aging, the intestine and the hypodermal cells are the primary energy storage tissues in *C. elegans* (*Mak, 2012*; *Mullaney and Ashrafi, 2009*).

A WormCat 2.0 (*Higgins et al., 2022*; *Holdorf et al., 2020*) enrichment analysis of the WaRGs (*Figure 5B*) showed overlapping and differing enrichments between up- and downregulated genes. For example, we noted stress-related gene enrichments in both classes of WaRGs, consistent with known PI3K/Akt and p38 MAPK functions in *C. elegans* and with the strong correlation between stress response and longevity (*Zhou et al., 2011*). Among the non-overlapping enrichments, we found that the upregulated set was enriched for metabolism and transmembrane transport genes (*Figure 5B*). We found that collagen and neuropeptide-related genes (*Figure 5B*) were disproportionally represented among the non-overlapping but downregulated set. We were particularly surprised by the former

because atRA was identified as a longevity modulator through induction of a collagen (*col-144::gfp*) reporter (*Statzer et al., 2021*) and some observations correlate longevity with collagen expression (*Ewald et al., 2015*; *Goyala and Ewald, 2023*). Separation of the collagen and collagen-related genes by type *Teuscher et al., 2019* demonstrated a general trend of atRA either not changing or downregulating collagen genes (*Figure 5—figure supplement 1*). For example, the cuticular collagens, of which *col-144* is a predicted member, and other core genes associated with the extracellular matrix (matrisome genes) were either unchanged in expression or downregulated (*Figure 5—figure supplement 1*). In contrast with the core-matrisome genes, the matrisome-associated category did include a number of upregulated genes in the ECM-regulator and ECM-affiliated subclasses.

We also analyzed the WaRGs from a metabolic perspective using the WormFlux Pathway enrichment tool (*Walker et al., 2021*). Among the 166 upregulated WaRGs, we documented an enrichment of sphingolipid metabolism (8/45 genes, $p_{enrichment}$ = 2.3e−07) (*Figure 5—figure supplement 2*) and iron metabolism (2/15 genes, $p_{enrichment}$ = 0.025) pathway genes, while the 487 downregulated WaRGs were enriched for fatty acid biosynthesis (6/24 genes, $p_{enrichment}$ = 0.00051), fatty acid degradation (2/8, p = 0.048), folate biosynthesis (2/8 genes, $p_{enrichment}$ = 0.048), and UGT enzyme (8/67, $p_{enrichment}$ = 0.0094) pathway genes. Interestingly, WormFlux analysis also suggests that genes related to the electron transport chain may be under-represented (0/88 genes, $p_{depletion}$ = 0.013) among the down-regulated WaRGs.

Because of the potential overlap of enriched gene categories with the functions of the IIS-PI3K/Akt and Nrf2-p38 MAPK pathways in *C. elegans*, we wanted to determine if the genes with the largest fold change in expression were among the known IIS and Nrf2 regulons. Focusing on genes with a significant absolute LFC >3, we observed that among the 24 most upregulated genes, 83% (20) have previously been observed to be regulated by the IIS pathway and 71% (17) have been observed to be regulated by the Nrf2 pathway (*Supplementary file 1b*). Interestingly, the four genes without a known connection to the IIS pathway appear non-random. For example, the most upregulated gene (*E02C12.10*, LFC = 9.9) is a member of a family of 31 paralogs in *C. elegans* predicted to have kinase-like activity (*Davis et al., 2022*; *Vilella et al., 2009*). Interestingly, the E02C12.10 gene, which clearly merits further investigation, was also identified as a significant contributor to survival of AMPK-deficient dauer larvae using a genome-wide RNAi screen (*Xie and Roy, 2012*). Two additional members of the 'most upregulated' gene set (E02C12.12 and E02C12.6) were also members of this gene family, in addition to 10 additional genes in the upregulated WaRGS, representing 32% of all family members and ~48% (10/21) of the family members detected in our dataset, a significant enrichment over the observed rate (1.3%, p < 0.0001). In contrast with the upregulated WaRGs, none of the family members were classified as downregulated WaRGs. The function of these genes is unknown, but the family is defined by a putative protein kinase domain and a nuclear hormone receptor-like structure (*Davis et al., 2022*) that suggests a potential for transduction of an atRA regulatory response. There were fewer downregulated genes with an LFC <−3, with only 12 genes reaching the threshold. Among those 12 genes, 100% have previously been shown to be regulated by both the IIS and Nrf2 pathways (*Supplementary file 1c*).

Overall, consistent with our genetic results, there is a clear footprint of atRA activity across a broad set of stress-response and longevity-related pathways, with some indication of novel activity as well.

## The HSF-1 transcription factor is an important effector of the overall atRA transcriptional response

To further dissect the transcriptional response to atRA in detail, we repeated our transcriptional analysis in several mutant backgrounds. Using *hsf-1(sy441)* mutants, we were able to identify mRNA from 13,737 genes (compared to 12,746 in N2). A comparison of transcriptional responses to atRA for all genes shows that there is a strong correlation between the N2 and *hsf-1(sy441)* expression changes ($R^2$ = 0.306, p < 0.0001) (*Figure 6A*). Using the same cutoffs that we used for our wildtype dataset to define WaRGs, we determined that the atRA regulon for *hsf-1(sy441)* animals (298/13737) is ~42% the size of wildtype (653/12,746), with half of the response unique to *hsf-1(sy441)* (*Figure 6B*). Comparing the 653 WaRGs from the general analysis with the subset identified in *hsf-1* mutants, ~96% (629/653) were detected in both datasets. Among the 470 downregulated WaRGs, only 14% (64/470) still meet the WaRG thresholds in the *hsf-1* background. Among the 158 upregulated WaRGs that were detectible in our *hsf-1(sy441)* dataset, half (79/158) of the genes still met the threshold for classification

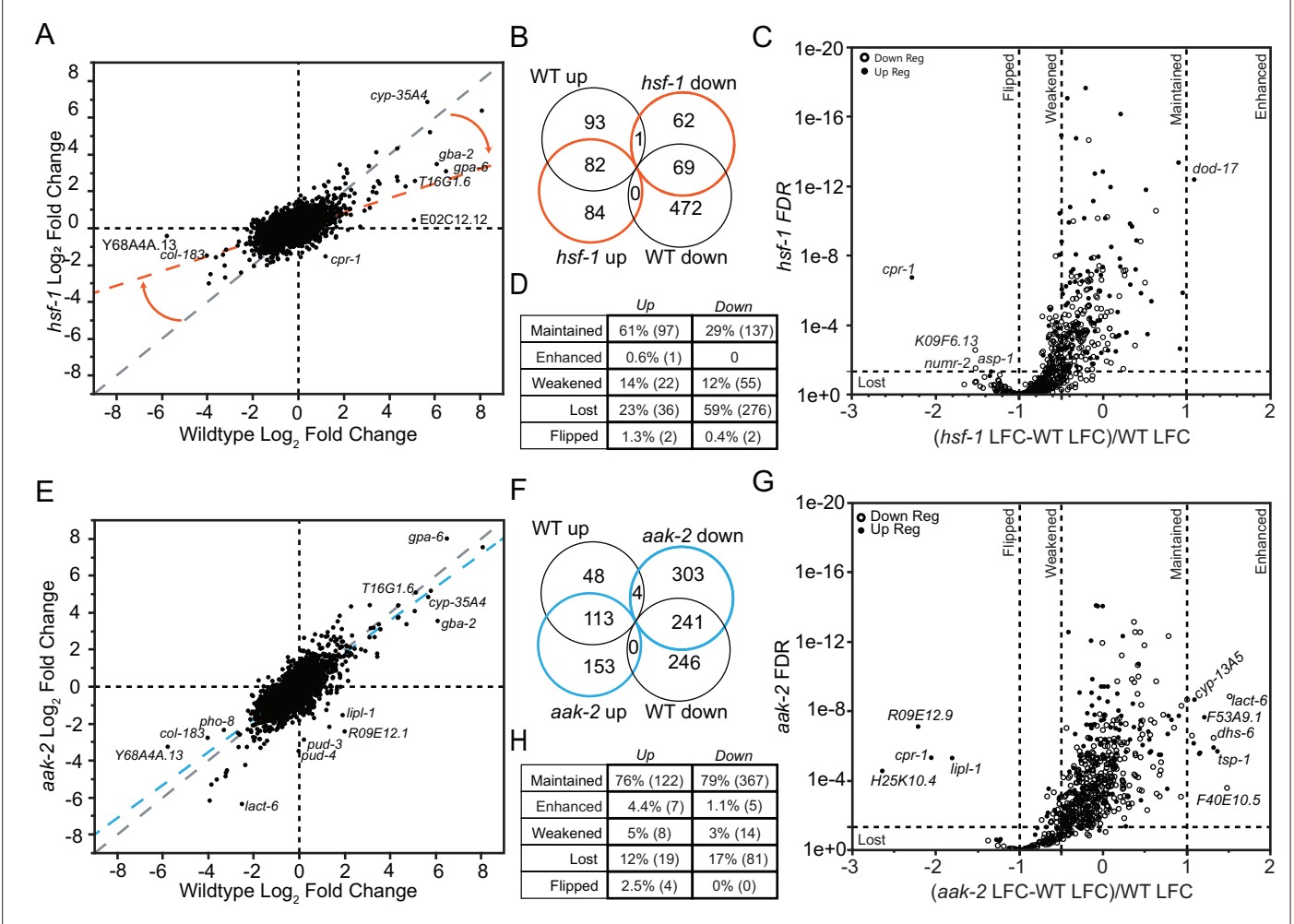

**Figure 6.** Analysis of the role of *hsf-1* and *aak-2* in atRA transcriptional response. (**A**) Comparison of DE genes with FDR <0.5 and |LFC| >1 in the *hsf-1* background to the changes observed in the wildtype background. The gray dashed line shows the expected relationship if the mutation had no effect on atRA response, while the orange dashed line shows the fit to the observed data. (**B**) Comparison of the genome-wide atRA-induced change in expression for all genes detected in both the N2 and *hsf-1(sy441)*. (**C**) Plot of the FDR for the WaRGs detected in the *hsf-1(sy441)* background by the difference in expression changes between WT and mutant background normalized to the change observed in WT animals. (**D**) Classification of WaRGs as maintaining, weakening, losing, or flipping their response in *hsf-1(sy441)* animals. (**E**) Comparison of the genome-wide atRA-induced change in expression for all genes detected in both the N2 and *aak-2(ok524)* datasets. The gray dashed line shows the expected relationship if the mutation had no effect on atRA response, while the blue dashed line shows the fit to the observed data. (**F**) Comparison of DE genes with FDR <0.5 and |LFC| >1 in the *aak-2(ok524)* background to the changes observed in the wildtype background. (**G**) Plot of the FDR for the WaRGs detected in the *aak-2(ok524)* background by the difference in expression change between WT and mutant background normalized to the change observed in WT animals. (**H**) Classification of WaRGs has maintaining, weakening, losing, or flipping their response in *aak-2(ok524)* animals.

The online version of this article includes the following figure supplement(s) for figure 6:

**Figure supplement 1.** The WormCat enrichment analysis for the WaRGs whose response was maintained, weakened, or lost in *hsf-1(sy441)* or *aak-2(ok524)* animals.

as a WaRG. The loss of differential expression could result from fewer genes changing expression or by a decrease in magnitude of the response that drops genes below our current threshold for defining WaRG genes. This potential for 'lost' regulation would be particularly skewed for genes whose expression change was near the absolute LFC = 1 threshold, where a negligible change could alter the categorization of the response. We therefore categorized the WaRG response as being maintained (0.5–2x WT response), weakened (<0.5X WT), lost (FDR >0.05), or flipped in the *hsf-1* background (**Figure 6C**). We found that a greater portion of the downregulated WaRG response was dependent on *hsf-1*, with only 29% of the downregulated response being maintained, compared to

61% of the upregulated response (*Figure 6D*). We then sought to determine if the lost and maintained WaRGs represented unique functions by performing an enrichment analysis using the WormCat analysis tool (*Figure 6—figure supplement 1*). We found that the maintained response was enriched for sphingolipid (6/44, p = 6.1e−06), sterol (6/59, p = 6.9e−04), and short chain dehydrogenase (6/42, p = 0.00011) metabolism genes. Additionally, the maintained response was also enriched for C-type lectin (10/256, p = 0.00029), CYP detoxification (10/82, p = 1.7e−07), and CUB pathogen (4/25, p = 0.004) stress-response genes. The same analysis of the lost WaRG response genes suggests an *hsf-1* dependence for atRA regulation of solute carrier (12/197, p = 0.000103), neuropeptide (10/139, p = 0.000196), and stress-response genes. Thus, *hsf-1* plays an important, but hardly absolute, role in mediating the atRA longevity response.

## Loss of AMPK results in both lost and gained transcriptional responses to atRA

To determine the role of AMPK in regulating the transcriptional response to atRA, we repeated our RNA-seq analysis in *aak-2(ok524)* mutants. A comparison of the transcriptional response to atRA for all genes shows that there is a strong correlation between the N2 and *aak-2(ok524)* data sets for expression changes ($R^2$ = 0.589, p < 0.0001) (*Figure 6E*). Using the same cutoffs that we used for our wildtype dataset to define WaRGs, we determined that the atRA regulon for *aak-2(ok524)* mutants (802/12,611) is actually larger than wildtype (653/12,746), with more than half (456/802) of the response unique to *aak-2(524)* (*Figure 6F*). We next sought to determine what portion of the wildtype atRA transcriptional response was lost in *aak-2* mutants. We therefore analyzed the 96% of the WaRGs (627/653) that were detected in our *aak-2* dataset and found that 76% (122/160) of the upregulated and 79% (367/467) of the downregulated WaRGs were similarly regulated in *aak-2* animals (*Figure 6G, H*). A WormCat 2.0 analysis (*Figure 6—figure supplement 1*) of the maintained atRA response demonstrated enrichment for secreted extracellular proteins (7/54, p = 0.004), collagen (12/184, p = 0.006), and solute carrier (18/197, p = 8.67e−7) genes. The maintained response was also enriched for sphingolipid (6/44, p = 1.5e−06) metabolism genes, suggesting that this metabolism category of changed expression observed in wildtype animals is upstream of *aak-2*. Additionally, there is an enrichment for caenacin (4/11, p = 0.007), and CUB pathogen (5/25, p = 0.009) stress-response genes. There was also an enrichment for ABC transmembrane transport (9/50, p = 82.9e−5) and cathepsin (5/22, p = 0.005) genes. We conclude that while *aak-2* is absolutely required for the longevity effects of atRA, *aak-2* is required for only a small proportion (~1/4) of the transcriptional response.

## AAK-2 functions downstream of HSF-1 in the transcriptional response to atRA

Given that HSF-1 and AAK-2 are both required for atRA lifespan extension, we sought to determine if HSF-1 and AAK-2 act in series or in parallel. Because a typical genetic analysis of longevity would not enable such a determination, we turned to the atRA transcriptional response (653 WaRGs) identified in wildtype animals. Compared to our datasets from *aak-2(ok524)* and *hsf-1(sy441)*, 610/653 WaRGs were detectible in both mutants. We therefore analyzed those 610 genes for patterns of transcriptional response. We observed that *aak-2(ok524)* mutants retained a larger portion of the response, with 84.4% (515/610) of the WaRGs still being differentially expressed (FDR <0.05) in the *aak-2(ok524)* mutants, while only 52.6% (315/610) were in *hsf-1(sy441)* animals.

We next sought to categorize the atRA response overlap between mutants. We first used our normalized LFCΔ-based classification of WaRGs (maintained, enhanced, lost, weakened, or flipped) to determine the relationship between regulation in *hsf-1* and *aak-2* backgrounds. We find that nearly 87% (85/98) of the lost response in *aak-2* was also lost in *hsf-1* animals, while nearly 90% of the response retained in *hsf-1(sy441)* animals was retained in *aak-2(ok524)* animals (208/232; *Supplementary file 1d*). These results are inconsistent with two parallel responses where we would expect (mostly) non-overlapping classes of regulated (lost) genes. In fact, we see a significant enrichment of overlap beyond the expected overlap for random regulation between *hsf-1* and *aak-2*. This suggests that HSF-1 and AAK-2 regulators act in series, with *hsf-1* upstream of *aak-2*, in the atRA pathway.

We then reanalyzed the WaRGs after subsetting based on response in the mutant backgrounds. We found no enrichments at the levels used in our previous analyses for the response lost in both genetic backgrounds. In contrast, we observed enrichments for iron, amino acid, and sphingolipid metabolism

among those genes whose response was maintained in both *hsf-1* and *aak-2* backgrounds. We interpret these changes to be either independent of the atRA longevity pathway, or upstream of *hsf-1* in the atRA longevity response. A similar analysis of the 212 WaRGs that were lost in *hsf-1(sy441)*, but retained in *aak-2*, showed an enrichment for fatty acid biosynthesis and UGT pathway genes. Interestingly, HSF1 has been implicated in regulating fatty acid biosynthesis in mammals (*Jin et al., 2011*), suggesting a potentially conserved *hsf-1* function that lies upstream of *aak-2* in the atRA longevity response.

## Discussion

The translation of the biology of aging to improvements in human health will require an extensive and varied set of interventions as candidates for clinical trials. Among these interventions, small drug-like chemical compounds, or actual approved drugs, are likely to feature in translation. There is now a 25-year-old history of experiments showing small molecule extension in lifespan in simple laboratory animals and a 15-year history of extending lifespan in laboratory mice. Identifying novel compounds that hold the potential to extend life, especially if they do so by increasing the overall period of healthy living (healthspan) and not just lifespan per se (*Crimmins, 2015*; *EbioMedicine, 2015*) is of import. While there have been some celebrated successes in this area, the chemical space explored to date for longevity interventions is small. Hence, the field is shifting toward a systematic appraisal of a more comprehensive set of target compounds. One potential method of accomplishing this goal is to use a broad collection of information on biological activity and structural characteristics of individual compounds to create a 'training set' that allows computational prediction of compound effects, thereby providing a means of prioritizing validation efforts in the face of many hundreds of thousands of potential options. Here we present a 'proof of concept' of this approach using a comprehensive, multi-species approach in *Caenorhabditis* nematodes via the CITP that draws upon a previously published set of compounds predicted to have positive effects on lifespan-related pathways (*Fuentealba et al., 2019*). Overall, focusing primarily on top-ranked and novel compounds, we find the mining of these predictions can be highly effective, with more than 31% of tested compounds leading to an increase in lifespan. When this list is augmented by additional predicted compounds previously tested by the CITP (and therefore not retested here), the prediction success rate stays very similar at 30% (*Table 1*). In comparison, several large-scale compound screens in *C. elegans* yielded much lower initial hit rates (<2%) and required multiple rounds of experimental validation to narrow down compounds to move forward into full lifespan assays (*Petrascheck et al., 2007*; *Lucanic et al., 2017*). The retest hit rate in both of these studies was less than 0.2%.

While most compounds tested here had relatively moderate effects (<15% increase in median lifespan), two interventions conferred large effects, including propranolol with a greater than 44% increase in median lifespan and all-trans retinoic acid (atRA), with a greater than 23% increase in median lifespan in *C. elegans*. The effects of both compounds were variable and were much reduced in related species *C. briggsae* and *C. tropicalis*, which has been a common feature of CITP tests for reasons that remain currently unknown (*Banse et al., 2024b*; *Banse et al., 2019*; *Lucanic et al., 2017*; *Onken et al., 2022*). Fortunately, *C. elegans* itself has been a reliable testing platform, including robust responses across a wide set of genetic backgrounds (*Banse et al., 2019*). Tests of the effect of propranolol directly on bacterial growth suggest that the increase in lifespan with that treatment might be caused by a dietary-restriction-like response in the nematodes, since growth of the bacteria that serve as their food source is inhibited under propranolol exposure. This effect deserves further investigation but was outside of the scope of the current project.

Of the sixteen compounds initially targeted, atRA emerged as the most interesting candidate, with positive effects on both lifespan and locomotory healthspan across diverse natural isolates of *C. elegans*. The positive effects of atRA have also been indicated by other studies, which generated a positive hit using a distinct approach involving the maintenance of collagen expression in adult *C. elegans* (*Statzer et al., 2021*). As such, atRA presented itself as an ideal candidate for using the power of nematode genetics to move from computational prediction to functional analysis. The fact that atRA is already an FDA-approved intervention for other indications makes it a particularly inviting compound. This observation is strengthened by the recent observation that the atRA precursor vitamin A can also enhance longevity in *C. elegans* and does so in a *skn-1*-dependent manner (*Sirakawin et al., 2024*).

## Putative atRA targets

Analysis of mutants in a number of key regulatory and stress-response systems treated with atRA suggests that atRA functions through the AKT-1 and AKT-2 kinases to affect conserved AMPK, Nrf2, and HSF1 pathways. Using a comprehensive RNA-seq approach with and without atRA treatment in both wildtype and mutant backgrounds suggests extensive remodeling of sphingolipid and fatty acid metabolic networks, both of which are known to modulate lifespan. While these data support a model for atRA affecting longevity through Akt and its downstream longevity transcription factors *hsf-1* and *skn-1*, the mechanism of initiation upstream of Akt remains unknown. One explanatory model of upstream initiation is suggested by our observation that atRA transcriptionally alters sphingolipid metabolism in *C. elegans*, which has also been seen in mammals (*Camdzic et al., 2023*; *Clarke et al., 2011*; *Kalén et al., 1992*; *Sun and Wang, 2021*). Sphingolipids are known to regulate developmental rate and lifespan in *C. elegans* (*Cutler et al., 2014*), and potentially in mammals as well (*Cutler and Mattson, 2001*). For example, remodeling the sphingolipid metabolism network through RNAi-induced reductions in *ttm-5* (dihydroceramide desaturase homologue), *W02F12.2* (neutral/acidic ceramidase homologue), *cgt-2* (glucosylceramide synthase homologue) or *K06A9.1* (neutral sphingomyelinase homolog), all result in lifespan extension. Additionally, genetic disruption of the ceramide synthases alters lifespan, with loss of *hyl-2* shortening and simultaneous loss of *hyl-1* and *lagr-1* extending *C. elegans* lifespan through a *skn-1*-dependent process (*Mosbech et al., 2013*). Additionally, the control of the relative ceramide and sphingomyelin levels by sphingomyelin synthases mediates crosstalk between DAF-16 and CREBH (*He et al., 2021*), which would have a significant impact on glucose and lipid metabolism, and therefore longevity. As such, the atRA-altered sphingolipid network observed in our RNA-seq data (*Figure 5B*; *Figure 5—figure supplement 2*) could have significant impacts on longevity.

The importance of sphingolipid regulation of metabolism, and ultimately lifespan, is not a unique feature of nematodes. The network has been proposed to function in mammals as a metabolic rheostat that uses the ratios of ceramide, ceramide-1P, sphingosine, and sphingosine-1P to determine metabolic regulatory response (*Summers et al., 2019*). This may be a mechanistic contributor to atRA longevity effects as there is known cross-talk between longevity pathways and ceramide/sphingolipids (*Jęśko et al., 2019*). Interestingly, ceramide and sphingolipid metabolism may provide a conserved functional connection to the observed relationship between atRA and protein kinase B/Akt function. In cell culture, treatment with atRA increases ceramide levels (*Kalén et al., 1992*), and cell-permeable ceramide inhibits Akt kinase activity (*Zhou et al., 1998*). Additionally, exogenous ceramide induces dephosphorylation and inhibition of Akt (*Zinda et al., 2001*). This functionality is known to be biologically relevant, as ceramide is a known negative regulator of insulin activity via regulation of Akt (*Hsieh et al., 2014*).

Considering these findings, a simple model consistent with our observations is that the application of atRA changes sphingolipid metabolism, which in turn induces a change in the functional state of *akt-1* and *akt-2*. Our observations that HSF-1, AAK-2, and SKN-1 are necessary for atRA longevity extension are easily understood within this model, as all three have been identified as potential direct targets of Akt regulation. How (and if) atRA directly regulates sphingolipid metabolism remains an open question. In cell culture, atRA induces growth arrest in many cell types, and that arrest is mediated through nSMase2 induction, which increases ceramide levels (*Clarke et al., 2011*). Additionally, the involvement of sphingosine kinases in atRA signaling has been demonstrated in K562 chronic myeloid leukemia cells (*Sun and Wang, 2021*), but the identity of the transcriptional effector remains unclear.

## Conservation of aging effects of atRA

The retinoids – atRA in particular – are broadly conserved regulators of transcription (*Amann et al., 2011*). In vertebrates, atRA functions in a broad range of biological activities, from development (*Duester, 2008*; *Niederreither and Dollé, 2008*), immune function (*Huang et al., 2018*), and memory and learning, to energy metabolism (*Zhang et al., 2015*). In mammals, some research suggests a potential role for atRA signaling in modulating aging. Among clinical aging studies of both natural and synthetic retinoids, the bulk of the research has been for aging and/or UV photoaging of skin. Among those studies, atRA is the most widely investigated retinoid and potentially the most potent (*Mukherjee et al., 2006*). Beyond skin phenotypes, studies in mouse models have also shown that

age-dependent decreases in atRA signaling result in poor performance on spatial learning and memory tasks, and dietary supplementation with atRA can ameliorate the age-related decreases in hippocampal long-term-memory potentiation and other brain functions (*Etchamendy et al., 2001*). The potential use of atRA as an intervention in age-related diseases of neurophysiology has not been ignored and is receiving significant attention as a therapeutic for Alzheimer's disease and related dementias (*Das et al., 2019*; *Lee et al., 2009*; *Szutowicz et al., 2015*).

While atRA may function as an anti-aging agent due to the phenotypic outcomes of application, the molecular mechanisms responsible for these activities are not fully understood. One possibility is that atRA functions as a high-affinity ligand for PPARβ/δ peroxisome proliferation-activated receptor, which is a master regulator of lipid metabolism and glucose homeostasis. Activation of PPARβ/δ increases lipid catabolism in adipose tissue and skeletal muscle to prevent obesity (*Kuri-Harcuch, 1982*; *Pairault et al., 1988*; *Sato et al., 1980*; *Schwarz et al., 1997*). Additionally, in an obese mouse model, treatment with atRA-induced PPARβ/δ and RAR regulated genes, correlating with weight loss and improved insulin responsiveness (*Berry and Noy, 2009*). Alternatively, atRA could be affecting longevity through effects on Akt proteins, as we observed and has been shown for other atRA phenotypes in mammals. Indeed, multiple pathways are likely to be engaged.

## Conclusions

We tested the hypothesis that using intersecting computational predictions can identify aging interventions at a high frequency in *Caenorhabditis* species. We found that using cross-validated computational predictions resulted in a high discovery rate (30%), which is compatible with screening across a dosage range using full lifespan analysis. The future success of computational prediction approaches should increase as AI methodologies are brought to bear on an ever-increasing body of research. In the example described above, computational predictions led to the identification of an endogenous signaling ligand that regulates metabolism and can be co-opted to extend life. Our study demonstrates the potential of metabolic manipulation for aging interventions and the benefits of computational predictions in prioritizing a compound screening.

# Materials and methods

**Key resources table**

| Reagent type (species) or resource | Designation | Source or reference | Identifiers | Additional information |
|---|---|---|---|---|
| Strain, strain background (*C. elegans*) | See strain list in Methods | This paper | | Available at *Caenorhabditis* Genetics Center |
| Strain, strain background (*C. briggsae*) | See strain list in Methods | This paper | | Available at *Caenorhabditis* Genetics Center |
| Strain, strain background (*C. tropicalis*) | See strain list in Methods | This paper | | Available at *Caenorhabditis* Genetics Center |
| Software, algorithm | Lifespan (R script) – ATRA | 10.6084/m9.figshare.26308177 | | |
| Software, algorithm | Lifespan (R script) – compound screen | 10.6084/m9.figshare.26308153 | | |
| Software, algorithm | Lifespan (R script) – propranolol PFA-killed OP50-1 | 10.6084/m9.figshare.26308159 | | |
| Software, algorithm | Lifespan (R script) – pathway mutants | 10.6084/m9.figshare.26308165 | | |
| Software, algorithm | Lifespan (R script) – ATRA automated lifespan | 10.6084/m9.figshare.26308186 | | |
| Software, algorithm | CeleST (R script) | 10.6084/m9.figshare.26308198 | | |
| Software, algorithm | Lifespan (R script) – *C. briggsae* and *C. tropicalis* | 10.6084/m9.figshare.26308171 | | |
| Software, algorithm | RNA-seq (R script) | 10.6084/m9.figshare.26314531 | | |

*Continued on next page*

*Continued*

| Reagent type (species) or resource | Designation | Source or reference | Identifiers | Additional information |
|---|---|---|---|---|
| Software, algorithm | Transcriptomic alignments and feature counts | 10.6084/m9.figshare.26314591 | | |
| Chemical compound/drug | Bortezomib | Sigma-Aldrich | | |
| Chemical compound/drug | Tretinoin | Sigma-Aldrich | | |
| Chemical compound/drug | Fisetin | Tocris Bioscience | | |
| Chemical compound/drug | Temsirolimus | Cayman Chemical | | |
| Chemical compound/drug | Everolimus | Cayman Chemical | | |
| Chemical compound/drug | Dasatinib | Cayman Chemical | | |
| Chemical compound/drug | Decitabine | Selleck Chemicals | | |
| Chemical compound/drug | Gefitinib | Sigma-Aldrich | | |
| Chemical compound/drug | Metoprolol | Sigma-Aldrich | | |
| Chemical compound/drug | Berberine | Cayman Chemical | | |
| Chemical compound/drug | Erlotinib | Cayman Chemical | | |
| Chemical compound/drug | Dexamethasone | Sigma-Aldrich | | |
| Chemical compound/drug | Aldosterone | Sigma-Aldrich | | |
| Chemical compound/drug | Propranolol | Sigma-Aldrich | | |
| Chemical compound/drug | Metoprolol | Sigma-Aldrich | | |

A detailed set of standard operating procedures is available online (*Caenorhabditis Intervention Testing Program, 2025*). The experimental details in brief are as follows:

### Caenorhabditis strains and maintenance

All Caenorhabditis strains were obtained from the *Caenorhabditis* Genetics Center: N2-PD1073 *Teterina et al., 2022*; *Yoshimura et al., 2019*; CF1038 daf-16(mu86) (*Lin et al., 1997*); DA1113 eat-2(ad1113) *Raizen et al., 1995*; IG10 tol-1(nr2033) *Pujol et al., 2001*; RB754 aak-2(ok524) *C. elegans Deletion Mutant Consortium, 2012*; PS3551 *hsf-1(sy441) Hajdu-Cronin et al., 2004*; *KU25 pmk-1(km25) Mizuno et al., 2004*; RB759 akt-1(ok525); and VC204 akt-2(ok393). Wild isolates in this study include *C. elegans* JU775 and MY16, *C. briggsae* AF16, ED3092, and HK104, and *C. tropicalis* JU1630, JU1373, and QG834. All strains were maintained on nematode growth medium plates seeded with *Escherichia coli* OP50-1 at 20°C. For experimental synchronization, cohorts were generated by timed egg lays (*Lucanic et al., 2017*).

### Compound treatment

Compound treatment was conducted as previously published (*Banse et al., 2019*; *Lucanic et al., 2017*). Compounds were obtained as a solid and dissolved in DMSO (dimethyl sulfoxide) or $H_2O$ to obtain a stock solution. The following compounds were used: temsirolimus (Cayman 11590), ritonavir (Sigma-Aldrich SML0491), thalidomide (Calbiochem 585970), arecoline (Cayman 13662), everolimus (Cayman 11597), temsirolimus (Cayman 11590), erlotinib (Cayman 10483), berberine (Cayman 10006427), dasatinib (Cayman 11498), propranolol (Sigma-Aldrich P0884), aldosterone (Sigma-Aldrich A9477), dexamethasone (Sigma-Aldrich D1756), gefitinib (Sigma-Aldrich SML1657), tretinoin (all-trans retinoic acid) (Sigma-Aldrich PHR1187), bortezomib (Sigma-Aldrich 5043140001), decitabine (Selleck Chemical S1200), fisetin (Tocris 5016), and metoprolol (Sigma-Aldrich M5391). DMSO stock solutions were mixed with water to form a working solution before being added to plates. In-plate concentrations were calculated by presuming the final volume to be equal to that of the volume of agar. For 50, 100, and 150 µM atRA plates, stock solutions formed precipitates at working solution concentrations, requiring working solutions to be prepared individually for each plate.

## Lifespan assays

Lifespan assays were performed as previously published (*Banse et al., 2019*; *Lucanic et al., 2017*). Briefly, worms were age-synchronized via timed egg lays and transferred to control or compound treated plates on days 1, 2, and 5 of adulthood (or day 4 for *C. tropicalis* strains), and once weekly thereafter until dead. All lifespans were conducted using 51 µM FUdR (5-fluoro-2′-deoxyuridine) to prevent progeny production (*Hosono, 1978*; *Lucanic et al., 2017*; *Mitchell et al., 1979*). For automated lifespan assays, worms were transferred to the Automated Lifespan Machines (Epson Perfection V800s) (*Abbott et al., 2020*) on day 5 (*C. elegans* and *C. briggsae*) or day 4 (*C. tropicalis*) of adulthood, at which point survival data was collected and analyzed using the Lifespan Machine software (https://github.com/nstroustrup/lifespan; *Stroustrup, 2022*; *Stroustrup et al., 2013*). For lifespans using RNAi feeding, RNAi plates (25 mg/l carbenicillin and 1 mM IPTG) were seeded using an RNase II deficient *E. coli* strain (HT115) harboring either the *skn-1* (T19E7.2) targeting or control L4440 plasmid from the Ahringer RNAi library (*Kamath and Ahringer, 2003*). Worms were transferred to RNAi plates at the L3/L4 stage before being transferred to compound-treated RNAi plates containing 51 µM FUdR on day 1 of adulthood. An additional transfer on day 3 of adulthood was also added for lifespans using RNAi. All lifespan assays were conducted at 20°C and 80% relative humidity with 50 animals per Petri plate. Analysis of propranolol effects in the presence of paraformaldehyde-treated bacteria was performed as published (*Beydoun et al., 2023*).

## CeleST health assays

*CeleST* health assays were performed as previously published (*Banse et al., 2024b*). In brief, animals were exposed to compound intervention during adulthood as described above until CeleST measurements were collected at two time points (adult days 6 and 12 for *C. elegans* and *C. tropicalis*, and days 8 and 16 for *C. briggsae*). For two biological replicates at each of the three CITP sites, 40 animals were tested per condition (age and compound or control) per strain. For full experimental protocols, see our online protocol (*Caenorhabditis Intervention Testing Program, 2022*). Eight different parameters (wave initiation rate, body wave number, asymmetry, stretch, curling, travel speed, brush stroke, and activity index; *Ibáñez-Ventoso et al., 2016*; *Restif et al., 2014*) were measured using the CeleST software and used to create a composite swimming score (*Banse et al., 2024b*).

## Statistical analysis

Statistical analyses for lifespan experiments were performed as previously described (*Lucanic et al., 2017*). In summary, we used a mixed-model approach where compound treatment was considered a fixed effect, and other potential variables were treated as random effects. Survival was analyzed both with generalized linear models using the lme4 (version 1.1.32) package (*Bates et al., 2015*), and a mixed-model Cox-Proportional Hazards (CPH) model using the coxme package (version 2.2-18.1) (*Therneau, 2020*) in the R statistical language (*R Development Core Team, 2021*). The effect of compound treatment was tested using CPH analysis within each strain to allow for each compound treatment replicate to be compared to its specific control in the randomized blocks design. Compound effects were analyzed as a planned comparison between the responses of individuals on the treatment in question and those on the appropriate treatment control. Hits were classified based on a significant *p*-value from the CPH model coupled with an increase in median lifespan. It should be noted that one compound, aldosterone at 50 µM, showed a significant decrease in the hazard estimate without an increase in median lifespan, and thus was not considered a hit.

Swimming behavior was analyzed using the composite score (described above) as the variable of interest in mixed effects general linear models built for each strain in R using the lme4 package (version 1.1.32) (*Bates et al., 2015*) as previously described (*Banse et al., 2024b*). Determination of significant age by compound interactions was made using the R car package (version 3.1-2) (*Fox and Weisberg, 2019*).

## Transcriptomic analysis

For RNA-sequencing, worms were synchronized and compound treated as described above. Four biological replicates of both atRA-treated (150 µM) and vehicle control worms were aged to day 4 of adulthood and collected in tandem (approximately 50 worms total per replicate). We selected this timepoint because it corresponded to the timepoint at which the Ewald study detected increased

*col-144p::GFP* that predicted longevity (*Statzer et al., 2021*). Worms were picked into 0.2 ml tubes each containing 50 µl of lysis buffer (45 µl elution buffer plus 5 µl proteinase K) and flash frozen with liquid nitrogen, then stored at –80°C until library prep. Libraries were prepared using the KAPA mRNA HyperPrep kit (KK8580 from Kapa Biosystems) as per the manufacturer's protocol except that the total volume was adjusted to ¼ per reaction. Final libraries were normalized by concentration and sequenced on an Illumina Novaseq 6000 with the SP 100 cycle (GC3F, University of Oregon).

Paired-end FASTQ files for all atRA-treated and DMSO control samples were aligned to the *C. elegans* WBcel235 (build 104) reference genome using the Subread package (version 2.0.2) (*Liao et al., 2013*). Uniquely mapped reads were assigned to *C. elegans* genes with Subread's feature-Counts program (*Liao et al., 2014*) using reversely stranded read counting. Subsequent filtering, normalization, and differential expression analysis were performed on each strain-specific dataset with the edgeR package (version 3.28.1) (*Robinson et al., 2010*), using R (version 3.6.2) (*R Development Core Team, 2021*). Lowly expressed genes were removed from each dataset; only genes that had at least 10 reads in at least four samples and a minimum total count of 15 reads across samples were retained. To remove composition biases between libraries, the library sizes were normalized using a trimmed mean of *M*-values (*Robinson and Oshlack, 2010*) between each pair of samples. A pair-wise expression analysis was performed on the transcriptomes of the treatment and control samples from each strain. Quasi-likelihood *F*-tests for treatment vs. control sample effect were carried out on fitted gene-wise negative binomial generalized log-linear models. p-values were corrected for false discovery using the Benjamini–Hochberg method.

## Acknowledgements

We acknowledge and thank Max Guo (NIA), Viviana Perez (previously at NIA), Tiziana Cogliati (NIA), and the members of the Phillips, Driscoll, and Lithgow labs for helpful discussions and the Barber lab (University of Oregon) for kindly providing equipment access and technical expertise. Some strains were provided by the CGC, which is funded by NIH Office of Research Infrastructure Programs (P40 OD010440). Some deletion mutants used in this work were generated by the International *C. elegans* Gene Knockout Consortium (*C. elegans* Gene Knockout Facility at the Oklahoma Medical Research Foundation, which is funded by the National Institutes of Health; and the *C. elegans* Reverse Genetics Core Facility at the University of British Columbia, which is funded by the Canadian Institute for Health Research, Genome Canada, Genome BC, the Michael Smith Foundation, and the National Institutes of Health). This work was directly supported by funding from National Institutes of Health grants (U01 AG045844, U01 AG045864, U01 AG045829, and U24 AG056052).

## Additional information

### Funding

| Funder | Grant reference number | Author |
| --- | --- | --- |
| National Institutes of Health | AG045844 | Gordon Lithgow |
| National Institutes of Health | AG045864 | Monica Driscoll |
| National Institutes of Health | AG045829 | Patrick C Phillips |
| National Institutes of Health | AG056052 | Patrick C Phillips |

The funders had no role in study design, data collection, and interpretation, or the decision to submit the work for publication.

### Author contributions

Stephen A Banse, Conceptualization, Formal analysis, Investigation, Visualization, Methodology, Writing - original draft, Writing - review and editing; Christine A Sedore, Data curation, Formal

analysis, Investigation, Visualization, Writing - original draft, Writing - review and editing; Anna Coleman-Hulbert, Conceptualization, Investigation, Writing - original draft, Project administration; Erik Johnson, Investigation, Writing - original draft; Brian Onken, Supervision, Investigation; David Hall, Yuhua Song, Haley C Osman, Jian Xue, Elena Basttistoni, Anna Foulger, Madhuri Achanta, Mustafa Sheikh, Theresa Fitzgibbon, Investigation; Erik Segerdell, Data curation, Formal analysis, Visualization; E Grace Jackson, Data curation, Formal analysis; Suhzen Guo, Conceptualization, Project administration; John H Willis, Conceptualization; Gavin C Woodruff, Conceptualization, Formal analysis; Monica Driscoll, Gordon Lithgow, Supervision, Funding acquisition, Writing - review and editing; Patrick C Phillips, Conceptualization, Supervision, Funding acquisition, Writing - original draft, Project administration, Writing - review and editing

### Author ORCIDs
Stephen A Banse ⬚ https://orcid.org/0000-0002-5540-4526
Christine A Sedore ⬚ https://orcid.org/0000-0002-9512-2806
Anna Coleman-Hulbert ⬚ https://orcid.org/0000-0001-8090-551X
John H Willis ⬚ https://orcid.org/0000-0002-4263-7347
Monica Driscoll ⬚ https://orcid.org/0000-0002-8751-7429
Gordon Lithgow ⬚ https://orcid.org/0000-0002-8953-3043
Patrick C Phillips ⬚ https://orcid.org/0000-0001-7271-342X

Reviewer #1 (Public review): https://doi.org/10.7554/eLife.104375.3.sa1
Reviewer #2 (Public review): https://doi.org/10.7554/eLife.104375.3.sa2
Reviewer #3 (Public review): https://doi.org/10.7554/eLife.104375.3.sa3
Author response https://doi.org/10.7554/eLife.104375.3.sa4

---

## Additional files

### Supplementary files
Supplementary file 1. Supplementary tables providing additional quantitative detail for the results. Tables containing (a) sources of experimental variation across trials, genes significantly (b) up- and (c) downregulated by atRA treatment, and (d) WaRG comparisons across *hsf-1* and *aak-2* gene expression analyses.

MDAR checklist

### Data availability
All data is available at https://citpaging.org/portal and at https://doi.org/10.6084/m9.figshare.c.7350250 (Caenorhabditis Intervention Testing Program, 2025). Additionally, transcriptomic data have been deposited in NCBI's Gene are accessible through the NCBI GEO database via accession number GSE272535 (https://www.ncbi.nlm.nih.gov/geo/query/acc.cgi?acc=GSE272535).

The following datasets were generated:

| Author(s) | Year | Dataset title | Dataset URL | Database and Identifier |
|---|---|---|---|---|
| Banse SA, Sedore CA, Coleman-Hulbert AL, Johnson E, Onken B, Hall D, Segerdell E, Jones EG, Song Y, Osman H, Xue J, Battistoni E, Guo S, Foulger AC, Achanta M, Sheikh M, Fitzgibbon T, Willis JH, Woodruff GC, Driscoll M, Lithgow G, Phillips PC | 2024 | Computer prediction and genetic analysis identifies retinoic acid modulation as a driver of conserved longevity pathways in genetically-diverse Caenorhabditis nematodes | https://www.ncbi.nlm.nih.gov/geo/query/acc.cgi?acc=GSE272535 | NCBI Gene Expression Omnibus, GSE272535 |

*Continued on next page*

*Continued*

| Author(s) | Year | Dataset title | Dataset URL | Database and Identifier |
|---|---|---|---|---|
| Caenorhabditis Intervention Testing Program | 2025 | Caenorhabditis Intervention Testing Program: Data, analysis, and SOPs for the all-trans retinoic acid manuscript | https://doi.org/10.6084/m9.figshare.c.7350250 | figshare, 10.6084/m9.figshare.c.7350250 |

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
