## [Editor Report · eLife Assessment]

This **important** study explores the power of computational methods to predict lifespan-extending small molecules, demonstrating that while these methods significantly increase hit rates, experimental validation remains essential. The study uses all-trans retinoic acid in *Caenorhabditis elegans* as a model, providing genetic and transcriptomic insights into its longevity effects. The data are **compelling** in describing a robust, computationally informed screening process for discovering compounds that extend lifespan in this species.

---

## [Referee Report · Reviewer #1 (Public review)]

Summary:

This study highlights the strengths of using predictive computational models to inform *C. elegans* screening studies of compounds' effects on aging and lifespan. The authors primarily focus on all-trans retinoic acid (atRA), one of the 5 compounds (out of 16 tested) that extended *C. elegans* lifespan in their experiments. They show that atRA has positive effects on *C. elegans* lifespan and age-related health, while it has more modest and inconsistent effects (i.e., some detrimental impacts) for *C. briggsae* and *C. tropicalis*. In genetic experiments designed to evaluate contributing mediators of lifespan extension with atRA exposure, it was found that 150 µM of atRA did not significantly extend lifespan in akt-1 or akt-2 loss-of-function mutants, nor in animals with loss of function of aak-2, or skn-1 (in which atRA had toxic effects); these genes appear to be required for atRA-mediated lifespan extension. hsf-1 and daf-16 loss-of-function mutants both had a modest but statistically significant lifespan extension with 150 µM of atRA, suggesting that these transcription factors may contribute towards mediating atRA lifespan extension, but that they are not individually required for some lifespan extension. RNAseq assessment of transcriptional changes in day 4 atRA-treated adult wild type worms revealed some interesting observations. Consistent with the study's genetic mutant lifespan observations, many of the atRA-regulated genes with the greatest fold-change differences are known regulated targets of daf-2 and/or skn-1 signaling pathways in *C. elegans*. hsf-1 loss-of-function mutants show a shifted atRA transcriptional response, revealing a dependence on hsf-1 for ~60% of the atRA-downregulated genes. On the other hand, RNAseq analysis in aak-2 loss-of-function mutants revealed that aak-2 is only required for less than a quarter of the atRA transcriptional response. All together, this study is a proof of the concept that computational models can help optimize *C. elegans* screening approaches that test compounds' effects on lifespan, and provides comprehensive transcriptomic and genetic insights into the lifespan-extending effects of all-trans retinoic acid (atRA).

Strengths:

A clearly described and well-justified account describes the approach used to prioritize and select compounds for screening, based on using the top candidates from a published list of computationally ranked compounds (Fuentealba et al., 2019) that were cross-referenced with other bioinformatics publications to predict anti-aging compounds, after de-selecting compounds previously evaluated in *C. elegans* as per the DrugAge database. 16 compounds were tested at 4-5 different concentrations to evaluate effects on *C. elegans* lifespan.

Robust experimental design was undertaken evaluating the lifespan effects of atRA, as it was tested on three strains each of *C. elegans*, *C. briggsae*, and *C. tropicalis*, with trial replication performed at three distinct laboratories. These observations extended beyond lifespan to include evaluations of health metrics related to swimming performance.

In-depth analyses of the RNAseq data of whole-worm transcriptional responses to atRA revealed interesting insights into regulator pathways and novel groups of genes that may be involved in mediating lifespan-extension effects (e.g., atRA-induced upregulation of sphingolipid metabolism genes, atRA-upregulation of genes in a poorly-characterized family of *C. elegans* paralogs predicted to have kinase-like activity, and disproportionate downregulation of collagen genes with atRA).

Weaknesses:

The authors' computational-based compound screening approach led to a ~30% prediction success rate for compounds that could extend the median lifespan of *C. elegans*. However, follow-up experiments on the top compounds highlighted the fact that some of these observed "successes" could be driven by indirect, confounding effects of these compounds on the bacterial food source, rather than direct beneficial effects on *C. elegans* physiology and lifespan. For instance, this appeared to be the case for the "top" hit of propranolol. Other compounds were not tested with metabolically inert or killed bacteria to preclude the possibility of bacteria-produced metabolites exerting observed effects; this might be a useful future direction to consider.

Transcriptomic analyses of atRA effects were extensive in this study, but discussions of potential non-transcriptional effects of key proposed regulators (such as AMPK) were limited. For instance, other outputs of aak-2/AMPK (non-transcriptional changes to metabolic balance, autophagy, etc.) might account for its requirement for mediating lifespan extension effects, since aak-2 was not required for a major proportion of atRA transcriptional responses.

Comments on revisions:

In their revisions, the authors resolved all of my initial recommendations, and I have no additional suggestions.

---

## [Referee Report · Reviewer #2 (Public review)]

Summary:

In this manuscript, Banse et al. experimentally validate the power of computational approaches that predict anti-aging molecules using the multi-species approach of the Caenorhabditis Intervention Testing Program (CITP). Filtering candidate molecules based on transcriptional profiles, ML models, literature searches, and the DrugAge database, they selected 16 compounds for testing. Of those, eight did not affect *C. elegans*' lifespan, three shortened it, and five extended *C. elegans*' lifespan, resulting in a hit rate of over 30%. Of those five, they then focused on all-trans-retinoic acid (atRA), a compound that has previously resulted in contradictory effects. The lifespan-extending effect of atRA was consistent in all *C. elegans* strains tested, was absent in *C. briggsae*, and a small effect was observed in some *C. tropicalis* strains. Similar results were obtained for measures of healthspan. The authors then investigated the mechanism of action of atRA and showed that it was only partially dependent on daf-16 but required akt-1, akt-2, skn-1, hsf-1, and, to some degree, pmk-1. The authors further investigate the downstream effects of atRA exposure by conducting RNAseq experiments in both wild-type and mutant animals to show that some, but surprisingly few, of the gene expression changes that are observed in wild-type animals are lost in the hsf-1 and aak-2 mutants

Strengths:

Overall, this study is well-conceived and executed as it investigates the effect of atRA across different concentrations, strains, and species, including life and health span. Revealing the variability between sites, assays, and the method used is a powerful aspect of this study. It will do a lot to dispel the nonsensical illusion that we can determine a per cent increase in lifespan to the precision of two floating point numbers.

An interesting and potentially important implication arises from this study. The computational selection of compounds was agnostic regarding strain or species differences and was predominantly based on observations made in mammalian systems. The hit rate calculated is based on the results of *C. elegans* and not on the molecules' effectiveness in Briggsae or Tropicalis. If it were, the hit rate would be much lower. How is that? It would suggest that ML models and transcriptional data obtained from mammals have a higher predictive value for *C. elegans* than for the other two species. This selectivity for *C. elegans* over *C. tropicalis* and *C. briggsae* seems both puzzling and unexpected. The predictions for longevity were based on the transcriptional data in cell lines. Would it be feasible to compare the mammalian data to the transcriptional data in Fig. 5 and see how well they match? While this is clear beyond the focus of this study, an implied prediction is that running RNAseqs for all these strains exposed to atRA would reveal that the transcriptional changes observed in the strains where it extends lifespan the most should match the mammalian data best. Otherwise, how could the mammalian datasets be used to predict the effects for *C. elegans* over *C. briggsae* or *C. tropicalis* have more predictive for one species than the other? There are a lot of IFs in this prediction, but such an experiment would reconsider and validate the basis on which the original predictions were made.

Weaknesses:

Many of the most upregulated genes, such as cyps and pgps are xenobiotic response genes upregulated in many transcriptional datasets from *C. elegans* drug studies. Their expression might be necessary to deal with atRA breakdown metabolites to prevent toxicity rather than confer longevity. Because atRA is very light sensitive and has toxicity of breakdown, metabolites may explain some of the differences observed with the lifespan of machine effects compared to standard assay practices. However, the authors provide a potential explanation for that observation.

Comments on revisions:

The authors have adequately addressed my concerns and the paper is suitable for publication.

---

## [Referee Report · Reviewer #3 (Public review)]

Summary:

In this study, Banse et al., demonstrate that combining computer prediction with genetic analysis in distinct *Caenorhabditis* species can streamline the discovery of aging interventions by taking advantage of the diverse pool of compounds that are currently available. They demonstrate that through careful prioritization of candidate compounds, they are able to accomplish a 30% positive hit rate for interventions that produce significant lifespan extensions. Within the positive hits, they focus on all-trans retinoic acid (atRA) and discover that it modulates lifespan through conserved longevity pathways such as AKT-1 and AKT-2 (and other conserved Akt-targets such as Nrf2/SKN-1 and HSF1/HSF-1) as well as through AAK-2, a conserved catalytic subunit of AMPK. To better understand the genetic mechanisms behind lifespan extension upon atRA treatment, the authors perform RNAseq experiments using a variety of genetic backgrounds for cross comparison and validation. Using this current state-of-the-art approach for studying gene expression, the authors determine that atRA treatment produces gene expression changes across a broad set of stress-response and longevity-related pathways. Overall, this study is important since it highlights the potential of combining traditional genetic analysis in the genetically tractable organism *C. elegans* with computational methods that will become even more powerful with the swift advancements being made in artificial intelligence. The study possesses both theoretical and practical implications not only in the field of aging, but also in related fields such as health and disease. Most of the claims in this study are supported by solid evidence, but the conclusions can be refined with a small set of additional experiments or re-analysis of data.

Strengths:

(1) The criteria for prioritizing compounds for screening are well-defined and is easy to replicate (Figure 1), even for scientists with limited experience in computational biology. The approach is also adaptable to other systems or model organisms.

(2) I commend the researchers for doing follow-up experiments with the compound propranolol to verify its effect of lifespan (Figure 2- figure supplement 2), given the observation that it affected the growth of OP50. To prevent false hits in the future, the reviewer recommends the use of inactivated OP50 for future experiments to remove this confounding variable.

(3) The sources of variation (Figure 3-figure supplement 2) are taken into account and demonstrates the need for advancing our understanding of the lifespan phenotype due to inter-individual variation.

(4) The addition of the *C. elegans* swim test in addition to the lifespan assays provides further evidence of atRA-induced improvement in longevity.

(5) The RNAseq approach was performed in a variety of genetic backgrounds, which allowed the authors to determine the relationship between AAK-2 and HSF-1 regulation of the retinoic acid pathway in *C. elegans*, specifically, that the former functions downstream of the latter.

Weaknesses:

(1) The authors demonstrate that atRA extends lifespan in a species-specific manner (Figure 3). Specifically, this extension only occurs in the species *C. elegans* yet, the title implies that atRA-induced lifespan extension occurs in different *Caenorhabditis* species when it is clearly not the case. While the authors state that failure to observe phenotypes in *C. briggsae* and *C. tropicalis* is a common feature of CITP tests, they do not speculate as to why this phenomenon occurs.

(2) There are discrepancies between the lifespan curves by hand (Figure 3-Figure supplement 1) and using the automated lifespan machine (Figure 3-supplement 3). Specifically, in the automated lifespan assays, there are drastic changes in the slope of the survival curve which do not occur in the manual assays and may be suggestive that confounding factors may still operate or produce additional variation in ALM experiments despite relatively well-controlled environmental conditions.

---

## [Author Response]

The following is the authors’ response to the original reviews.

**Reviewer #1 (Public review):**
Summary:This study highlights the strengths of using predictive computational models to inform *C. elegans* screening studies of compounds' eCects on aging and lifespan. The authors primarily focus on all-trans retinoic acid (atRA), one of the 5 compounds (out of 16 tested) that extended *C. elegans* lifespan in their experiments. They show that atRA has positive eCects on *C. elegans* lifespan and age-related health, while it has more modest and inconsistent eCects (i.e., some detrimental impacts) for *C. briggsae* and *C. tropicalis*. In genetic experiments designed to evaluate contributing mediators of lifespan extension with atRA exposure, it was found that 150 µM of atRA did not significantly extend lifespan in akt1 or akt-2 loss-of-function mutants, nor in animals with loss of function of aak-2, or skn-1 (in which atRA had toxic eCects); these genes appear to be required for atRA-mediated lifespan extension. hsf-1 and daf-16 loss-of-function mutants both had a modest but statistically significant lifespan extension with 150 µM of atRA, suggesting that these transcription factors may contribute towards mediating atRA lifespan extension, but that they are not individually required for some lifespan extension. RNAseq assessment of transcriptional changes in day 4 atRA-treated adult wild-type worms revealed some interesting observations. Consistent with the study's genetic mutant lifespan observations, many of the atRA-regulated genes with the greatest fold-change diCerences are known regulated targets of daf-2 and/or skn-1 signaling pathways in *C. elegans*. hsf-1 loss-offunction mutants show a shifted atRA transcriptional response, revealing a dependence on hsf-1 for ~60% of the atRA-downregulated genes. On the other hand, RNAseq analysis in aak-2 loss-of-function mutants revealed that aak-2 is only required for less than a quarter of the atRA transcriptional response. All together, this study is proof of the concept that computational models can help optimize *C. elegans* screening approaches that test compounds' eCects on lifespan, and provide comprehensive transcriptomic and genetic insights into the lifespan-extending eCects of all-trans retinoic acid (atRA).Strengths:(1) A clearly described and well-justified account describes the approach used to prioritize and select compounds for screening, based on using the top candidates from a published list of computationally ranked compounds (Fuentealba et al., 2019) that were crossreferenced with other bioinformatics publications to predict anti-aging compounds, after de-selecting compounds previously evaluated in *C. elegans* as per the DrugAge database. 16 compounds were tested at 4-5 diCerent concentrations to evaluate eCects on *C. elegans* lifespan.(2) Robust experimental design was undertaken evaluating the lifespan eCects of atRA, asit was tested on three strains each of *C. elegans*, *C. briggsae,* and *C. tropicalis*, with trial replication performed at three distinct laboratories. These observations extended beyond lifespan to include evaluations of health metrics related to swimming performance.(3) In-depth analyses of the RNAseq data of whole-worm transcriptional responses to atRA revealed interesting insights into regulator pathways and novel groups of genes that may be involved in mediating lifespan-extension eCects (e.g., atRA-induced upregulation of sphingolipid metabolism genes, atRA-upregulation of genes in a poorly-characterized family of *C. elegans* paralogs predicted to have kinase-like activity, and disproportionate downregulation of collagen genes with atRA).

We thank the reviewer for highlighting the strengths of our paper.

Weaknesses:(1) The authors' computational-based compound screening approach led to a ~30% prediction success rate for compounds that could extend the median lifespan of *C. elegans*. However, follow-up experiments on the top compounds highlighted the fact that some of these observed "successes" could be driven by indirect, confounding eCects of these compounds on the bacterial food source, rather than direct beneficial eCects on *C. elegans* physiology and lifespan. For instance, this appeared to be the case for the "top" hit of propranolol; other compounds were not tested with metabolically inert or killed bacteria. In addition, there are no comparative metrics provided to compare this study's ~30% success rate to screening approaches that do not use computational predictions.

We do test whether compounds have a direct e:ect on bacterial growth. We have the text to clarify that fact. There may be potential lifespan e:ects from atRA due to changes in bacterial metabolites, however exploring that more fully is beyond the scope of the current work.

We very much appreciate the question regarding relative success. An appropriate benchmark for “hit rate” is perhaps best provided by Petrascheck, Ye & Buck (2007), who conducted a large-scale screen of 88,000 compounds for e:ects on adult lifespan in *C. elegans*. They found an initial screening hit rate of 1.2% (1083/88000), which were then retested for a verified hit rate of 0.13% (115/88000), with a retest failure rate of 89% (968/1083). Similarly, Lucanic et al. (2016) screened 30,000 compounds, with an initial hit rate of approximately 1.7% (~500/30000), or these 180 were selected for retesting, resulting in a final verified hit rate of 0.19% (57/29680), which is comparable to the Petrascheck et al. result. The text in the discussion has been modified to include these studies.

(2)Transcriptomic analyses of atRA eCects were extensive in this study, but evaluations and discussions of non-transcriptional eCects of key proposed regulators (such as AMPK) were limited. For instance, non-transcriptional eCects of aak-2/AMPK might account for its requirement for mediating lifespan extension eCects, since aak-2 was not required for a major proportion of atRA transcriptional responses.

We naturally agree with the reviewer that non-transcriptional e:ects are possible and well worth pursuing in future work. However, these e:ects will still show within our study, as any upstream non-transcriptional e:ects are likely to reveal themselves in downstream transcriptional changes, as measured here.

**Reviewer #2 (Public review):**
Summary:In this manuscript, Banse et al. experimentally validate the power of computational approaches that predict anti-aging molecules using the multi-species approach of the Caenorhabditis Intervention Testing Program (CITP). Filtering candidate molecules based on transcriptional profiles, ML models, literature searches, and the DrugAge database, they selected 16 compounds for testing. Of those, eight did not aCect *C. elegan*'s lifespan, three shortened it, and five extended *C. elegan*'s lifespan, resulting in a hit rate of over 30%. Of those five, they then focused on all-trans-retinoic acid (atRA), a compound that has previously resulted in contradictory eCects. The lifespan-extending eCect of atRA was consistent in all *C. elegans* strains tested, was absent in *C. briggsae*, and a small eCect was observed in some *C. tropicalis* strains. Similar results were obtained for measures of healthspan. The authors then investigated the mechanism of action of atRA and showed that it was only partially dependent on daf-16 but required akt-1, akt-2, skn-1, hsf-1, and, to some degree, pmk-1. The authors further investigate the downstream eCects of atRA exposure by conducting RNAseq experiments in both wild-type and mutant animals to show that some, but surprisingly few, of the gene expression changes that are observed in wild-type animals are lost in the hsf-1 and aak-2 mutants.Strengths:Overall, this study is well conceived and executed as it investigates the eCect of atRA across diCerent concentrations, strains, and species, including life and health span. Revealing the variability between sites, assays, and the method used is a powerful aspect of this study. It will do a lot to dispel the nonsensical illusion that we can determine a percent increase in lifespan to the precision of two floating point numbers.An interesting and potentially important implication arises from this study. The computational selection of compounds was agnostic regarding strain or species diCerences and was predominantly based on observations made in mammalian systems. The hit rate calculated is based on the results of *C. elegans* and not on the molecules' eCectiveness in Briggsae or Tropicalis. If it were, the hit rate would be much lower. How is that? It would suggest that ML models and transcriptional data obtained from mammals have a higher predictive value for *C. elegans* than for the other two species. This selectivity for *C. elegans* over *C. tropicalis* and *C. Briggsae* seems both puzzling and unexpected. The predictions for longevity were based on the transcriptional data in cell lines.

This is a common observation in the CITP for which we do not currently have a satisfying explanation. For whatever reason, *C. elegans* is much more responsive to compounds than other species, much like it is more responsive to RNAi and other environmental interventions. It may be less active in detoxifying external agents than the other species, although this is just speculation at the moment. We continue to investigate this question, but that work is beyond the scope of the present paper.

Would it be feasible to compare the mammalian data to the transcriptional data in Figure 5 and see how well they match? While this is clear beyond the focus of this study, an implied prediction is that running RNAseqs for all these strains exposed to atRA would reveal that the transcriptional changes observed in the strains where it extends lifespan the most should match the mammalian data best. Otherwise, how could the mammalian datasets be used to predict the eCects of *C. elegans* over *C. Briggsae* or *C. Tropicalis* have more predictive for one species than the other? There are a lot of IFs in this prediction, but such an experiment would reconsider and validate the basis on which the original predictions were made.

These questions are worth pursuing in the future but are beyond the scope of the current work.

Weaknesses:Many of the most upregulated genes, such as cyps and pgps are xenobiotic response genes upregulated in many transcriptional datasets from *C. elegans* drug studies. Their expression might be necessary to deal with atRA breakdown metabolites to prevent toxicity rather than confer longevity. Because atRA is very light sensitive and has toxicity of breakdown, metabolites may explain some of the diCerences observed with the lifespan of machine eCects compared to standard assay practices.

This is certainly a possibility, although we often observe longer lifespans on the ALM, perhaps because they themselves are stressful, thereby providing a more sensitive background environment for detecting positive stress response modulators.

**Reviewer #3 (Public review):**
Summary:In this study, Banse et al., demonstrate that combining computer prediction with genetic analysis in distinct Caenorhabditis species can streamline the discovery of aging interventions by taking advantage of the diverse pool of compounds that are currently available. They demonstrate that through careful prioritization of candidate compounds, they are able to accomplish a 30% positive hit rate for interventions that produce significant lifespan extensions. Within the positive hits, they focus on all-trans retinoic acid (atRA) and discover that it modulates lifespan through conserved longevity pathways such as AKT-1 and AKT-2 (and other conserved Akt-targets such as Nrf2/SKN-1 and HSF1/HSF-1) as well as through AAK-2, a conserved catalytic subunit of AMPK. To better understand the genetic mechanisms behind lifespan extension upon atRA treatment, the authors perform RNAseq experiments using a variety of genetic backgrounds for cross-comparison and validation. Using this current state-of-the-art approach for studying gene expression, the authors determine that atRA treatment produces gene expression changes across a broad set of stress-response and longevity-related pathways. Overall, this study is important since it highlights the potential of combining traditional genetic analysis in the genetically tractable organism *C. elegans* with computational methods that will become even more powerful with the swift advancements being made in artificial intelligence. The study possesses both theoretical and practical implications not only in the field of aging but also in related fields such as health and disease. Most of the claims in this study are supported by solid evidence, but the conclusions can be refined with a small set of additional experiments or re-analysis of data.Strengths:(1) The criteria for prioritizing compounds for screening are well-defined and easy to replicate (Figure 1), even for scientists with limited experience in computational biology. The approach is also adaptable to other systems or model organisms.(2) I commend the researchers for doing follow-up experiments with the compound propranolol to verify its eCect on lifespan (Figure 2 Supplement 2), given the observation that it aCected the growth of OP50. To prevent false hits in the future, the reviewer recommends the use of inactivated OP50 for future experiments to remove this confounding variable.(3) The sources of variation (Figure 3, Figure Supplement 2) are taken into account and demonstrate the need for advancing our understanding of the lifespan phenotype due to inter-individual variation.(4) The addition of the *C. elegans* swim test in addition to the lifespan assays provides further evidence of atRA-induced improvement in longevity.(5) The RNAseq approach was performed in a variety of genetic backgrounds, which allowed the authors to determine the relationship between AAK-2 and HSF-1 regulation of the retinoic acid pathway in *C. elegans*, specifically, that the former functions downstream of the latter.

We thank the reviewer for highlighting these strengths.

Weaknesses:(1) The filtering of compounds for testing using the DrugAge database requires that the database is consistently updated. In this particular case, even though atRA does not appear in the database, the authors themselves cite literature that has already demonstrated atRA-induced lifespan extension, which should have precluded this compound from the analysis in the first place.

As often happens in science, this work was initiated before Statzer et al. (2021) was published. As such, it is included in the test set.

(2) The threshold for determining positive hits is arbitrary, and in this case, a 30% positive hit rate was observed when the threshold is set to a lifespan extension of around 5% based on Figure 1B (the authors fail to explicitly state the cut-oC for what is considered a positive hit).

Any compound that statistically increases lifespan is considered a positive hit by the CITP. The CITP in general is powered to detect minimum e:ect sizes of 5%.

(3) The authors demonstrate that atRA extends lifespan in a species-specific manner (Figure 3). Specifically, this extension only occurs in the species *C. elegans* yet, the title implies that atRA-induced lifespan extension occurs in diCerent Caenorhabditis species when it is clearly not the case. While the authors state that failure to observe phenotypes in *C. briggsae* and *C. tropicalis* is a common feature of CITP tests, they do not speculate as to why this phenomenon occurs.

Please see the comment above.

(4) There are discrepancies between the lifespan curves by hand (Figure 3 Figure Supplement 1) and using the automated lifespan machine (Figure 3 Supplement 3). Specifically, in the automated lifespan assays, there are drastic changes in the slope of the survival curve which do not occur in the manual assays. This may be due to improper filtering of non-worm objects, improper annotation of death times, or improper distribution of plates in each scanner.

Our storyboarding SOP ensures that discrepancies in the shape of the curve are unlikely to be due to annotation errors. We check every page of the storyboard by hand, so all non-worm objects are excluded. Furthermore, the first and last ~10% of deaths are checked by hand (as we observed that these time points are the most likely to be wrongly called by the software), with a few deaths chosen at random from the middle to ensure that the software is calling death times accurately. If we find a high amount of inaccurately called deaths, the entire plate is annotated by hand. For this specific experiment, 18% of the total deaths were hand annotated. Plates are randomly distributed across each scanner in an e:ort to prevent bias. As noted above, it does appear that the ALM environment and the “by hand” environment are somewhat di:erent.

(5) The authors miss an opportunity to determine whether the lifespan extension phenotype attributed to the retinoic acid pathway is mostly transcriptional in nature or whether some of it is post-transcriptional. The authors even state "that while aak-2 is absolutely required for the longevity eCects of atRA, aak-2 is required only for a small proportion (~1/4) of the transcriptional response", suggesting that some of the eCects are post-transcriptional. Further information could have been obtained had the authors also performed RNAseq analysis on the tol-1 mutant which exhibited an enhanced response to atRA compared to wild-type animals, and comparing the magnitude of gene expression changes between the tol-1 mutant and all other genetic backgrounds for which RNAseq was performed.
**Reviewer #1 (Recommendations for the authors):**
(1) Will the raw RNA-seq data be publicly deposited? Please clarify. This would strengthen the value of the study.

All data is available. We have clarified this in the text.

(2) Since all-trans retinoic acid is a metabolite of vitamin A, it seems important to include a discussion of and reference to the recent study SKN-1/NRF2 upregulation by vitamin A is conserved from nematodes to mammals and is critical for lifespan extension in *Caenorhabditis elegans* (Sirakawin et al Cell Reports 2024). Sirakawin et al include data that corroborates and expands on the findings of the current study, including the observation that vitamin A reduces whole-body lipid deposition (agrees with some of the transcriptional findings in the current study); that vitamin A protects against oxidative stress; that vitamin A elevates expression of gst-5, skn-1, and pmk-1; and that loss-offunction mutation of skn-1 has similar eCects to the current study, in terms of suppressing lifespan-extending eCects of vitamin A. In addition, adding some discussion of oxidative stress would strengthen this work, in light of widespread perceptions of the antioxidant properties of vitamin A (and its metabolites).

Thank you for this suggestion. We have added this citation to the discussion.

(3) Minor typo: Lines 341-342 - After a sentence that contains the phrase "collagen and neuropeptide related genes", the next sentence uses the term "the latter" in reference to the collagen genes (should be "the former").

Edited in text.

(4) Minor correction: In Figure 6, the information in the figure legend is swapped for figure panels (A and B).

Edited in figure caption.

(5) To me, the subtitle heading "Loss of AMPK leads to a unique transcriptional profile in response to atRA treatment" (Line 403) is misleading, considering the contents of the text in that section, and the data presented in Figure 6.

We have altered this heading to reflect this comment.

**Reviewer #2 (Recommendations for the authors):**
Using diCerent colors for the diCerent testing sites would make Figure 3 more readable.

Edited so that each lab is represented by a di:erent shade of green.

**Reviewer #3 (Recommendations for the authors):**
It would be interesting to investigate the eCect of even higher concentrations of atRA as it has been reported that atRA accumulation is associated with deleterious phenotypes in mice (Snyder et al., 2020, FASEB J).

We tested the highest concentration (150 uM) based on the solubility of the compound using our standardized plate treatment protocol, so we are unable to test higher concentrations.

A good first guess for a downstream retinoid receptor is nhr-23 which is the homolog of the vertebrate ROR genes. Stehlin-Gaon et al. (2003, Nat Struct Mol Biol) have shown that atRA is a ligand for the orphan nuclear receptor RORβ. It might be interesting to study the eCects of atRA on an nhr-23::AID (auxin inducible degron) background. This would allow you to circumvent the developmental phenotypes as a result of nhr-23 knockdown. Patrick/Stephen

A few notes on the text/figures:

Line 342: I believe the authors meant "former" instead of "latter".

Corrected in text.

Line 346: Can you also highlight col-144 in Fig. 5 S1?

This is not really feasible, as it is in the cluster near the where the axes meet (red arrow).

Line 400: CUB pathogen - based on Figure 6 Supp 1, this occurs in aak-2 and not in hsf-1.

Great catch by the reviewer. We have updated the figure with the correct information.

Line 414: hedgehog-like signaling - occurs in hsf-1 instead of aak-2. Similar inconsistencies occur in lines 415 (sterol), 417 (C-type lectin), and 418 (unassigned pathogens)

We have updated the text to eliminate potential conflicts/confusion in the presentation here.

Line 434: I believe the authors meant Figure "6" instead of "7"

Edited in text.

Line 475: Is it "fifteen" or "sixteen" compounds initially targeted?

Edited in text.

Can you please include the population sizes for the lifespan assays if not yet included in the detailed protocol to be published in FigShare (to which I currently do not have access to)?

Added “50 animals per petri plate” to Lifespan Assay methods section; additionally, all sample sizes are included as a summary tab in each dataset on figshare.com (10.6084/m9.figshare.c.6320690).